# Variational State-Space Models for Localisation and Dense 3D Mapping in 6 DoF

**Atanas Mirchev**     **Baris Kayalibay**     **Patrick van der Smagt**     **Justin Bayer**

Machine Learning Research Lab, Volkswagen Group, Munich, Germany
{atanas.mirchev,bkayalibay,bayerj}@argmax.ai

## Abstract

We solve the problem of 6-DoF localisation and 3D dense reconstruction in spatial environments as approximate Bayesian inference in a deep state-space model. Our approach leverages both learning and domain knowledge from multiple-view geometry and rigid-body dynamics. This results in an expressive predictive model of the world, often missing in current state-of-the-art visual SLAM solutions. The combination of variational inference, neural networks and a differentiable raycaster ensures that our model is amenable to end-to-end gradient-based optimisation. We evaluate our approach on realistic unmanned aerial vehicle flight data, nearing the performance of state-of-the-art visual-inertial odometry systems. We demonstrate the applicability of the model to generative prediction and planning.

## 1 Introduction

We address the problem of learning representations of spatial environments, perceived through RGB-D and inertial sensors, such as in mobile robots, vehicles or drones. Deep sequential generative models are appealing, as a wide range of inference techniques such as state estimation, system identification, uncertainty quantification and prediction is offered under the same framework (Curi et al., 2020; Karl et al., 2017a; Chung et al., 2015). They can serve as so-called *world models* or environment simulators (Chiappa et al., 2017; Ha & Schmidhuber, 2018), which have shown impressive performance on a variety of simulated control tasks due to their predictive capability. Nonetheless, learning such models from realistic spatial data and dynamics has not been demonstrated. Existing spatial generative representations are limited to simulated 2D and 2.5D environments (Fraccaro et al., 2018).

On the other hand, the state estimation problem in spatial environments—SLAM—has been solved in a variety of real-world settings, including cases with real-time constraints and on embedded hardware (Cadena et al., 2016; Engel et al., 2018; Qin et al., 2018; Mur-Artal & Tardós, 2017). While modern visual SLAM systems provide high inference accuracy, they lack a predictive distribution, which is a prerequisite for downstream perception–control loops.

Our approach scales the above deep sequential generative models to real-world spatial environments. To that end, we integrate assumptions from multiple-view geometry and rigid-body dynamics commonly used in modern SLAM systems. With that, our model maintains the favourable properties of generative modelling and enables prediction. We use the recently published approach of Mirchev et al. (2019) as a starting point, in which a variational state-space model, called DVBF-LM, is extended with a spatial map and an attention mechanism. Our contributions are as follows:

- We use multiple-view geometry to formulate and integrate a differentiable raycaster, an attention model and a volumetric map.
- We show how to integrate rigid-body dynamics into the learning of the model.
- We demonstrate the successful use of variational inference for solving direct dense SLAM for the first time, obtaining performance close to that of state-of-the-art localisation methods.
- We demonstrate strong predictive performance using the learned model, by generating spatially-consistent real-world drone-flight data enriched with realistic visuals.
- We demonstrate the model's applicability to downstream control tasks by estimating the cost-to-go for a collision scenario.

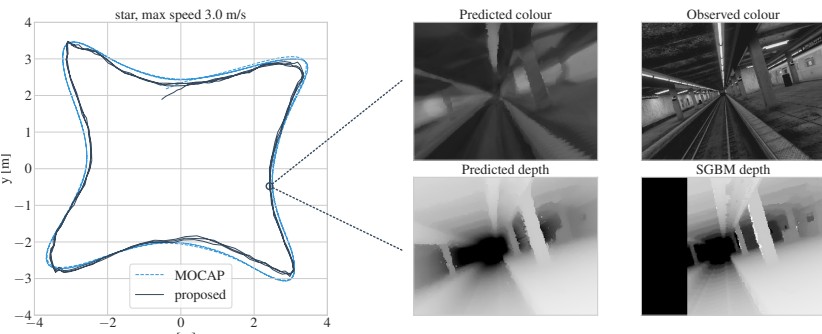

Figure 1: Illustration of the proposed quadcopter localisation and dense mapping. Left: top-down view of the localisaton estimate. Right: generative depth and colour reconstructions for one time step.

The contributions allow the reformulated model to tackle realistic RGB-D scenarios with 6 DoF.

## 2    RELATED WORK

**Generative models for spatial environments**    GTM-SM (Fraccaro et al., 2018) focuses on long-term predictions with a non-metric deterministic external memory. Chaplot et al. (2018) formulate an end-to-end learning model for active global localisation, filtering with a likelihood update predicted by a neural network. The agent can turn in four directions and move on a plane, perceiving images of the environment. VAST (Corneil et al., 2018) assumes a discrete state space for a generative model applied to the 2.5D Vizdoom environment. Whittington et al. (2018) model agents moving on a 2D grid with latent neurologically-inspired grid and place cells. Other works propose end-to-end learnable generative scene models (Eslami et al., 2018; Engelcke et al., 2020), without considering the agent dynamics. Like in the above, we put major emphasis on the generative predictive distribution of our model. With it, the agent can imagine the consequences of its future actions, a prerequisite for data-efficient model-based control (Chua et al., 2018; Hafner et al., 2019a;b; Becker-Ehmck et al., 2020). However, the aforementioned deep generative spatial models have only been applied on simulated 2D, 2.5D (movement restricted to a plane) and very simplified 3D environments.

A major challenge when scaling to the real world is to ensure that the learned components, and in turn the generative predictions, generalise to observed but yet unvisited places. Gregor et al. (2019) highlight another problem, that of long-term consistency when predicting ahead, and address it by learning with overshooting. In contrast, our method resolves these issues by injecting a sufficient amount of domain knowledge, without limiting the flexibility w. r. t. learning. To this end, we begin by sharing the probabilistic factorisation of DVBF-LM (Mirchev et al., 2019), a deep generative model that addresses the tasks of localisation, mapping, navigation and exploration in 2D. We then redefine the map, the attention, the states, the generation of observations and the overall inference, allowing for real-world 3D modelling and priming our method for data-efficient online inference in the future. We discuss why these changes are necessary more thoroughly in appendix A.

**Combining learning and spatial domain knowledge**    Fully-learned spatial models with an explicit memory component have been studied by Parisotto & Salakhutdinov (2018); Zhang et al. (2017); Oh et al. (2016). Further relying on geometric knowledge, Tang & Tan (2019) propose learning through the whole bundle adjustment optimisation, formulated on CNN feature maps of the observed images. Czarnowski et al. (2020) define a SLAM system based on learned latent feature codes of depth images, a continuation of the works by Zhi et al. (2019); Bloesch et al. (2018). Factor-graph maximum a posteriori optimisation is then conducted, substituting the observations for their respective low-dimensional codes, leading to point estimates of the individual geometry of $N$ keyframes and the agent poses over time. Wei et al. (2020) maintain cost volumes (Newcombe et al., 2011) for discretised poses and depth, and let a 3D CNN learn how to predict the correct geometry and pose estimates from them. Depth cost volumes are also used by Zhou et al. (2018) in learning to predict depth and odometry with neural networks. In the work by Yang et al. (2020), networks that predict odometry and depth are combined with DSO, leading to a SLAM system that utilises learning to its

advantage. Jatavallabhula et al. (2019) investigate differentiable SLAM, treating odometry estimation and mapping separately. The considered rendering gradients in that method are from the fused map to the observations, which is the opposite of the gradient paths used for learning in our work.

As in our approach, the geometric assumptions in the majority of these works allow the systems to generalise more easily to unseen cases and real-world data. What distinguishes our method is an explicit generative model, able to predict the agent movement and observations in the future. Additionally, our approach is fully-probabilistic, maintaining complete distributions over the variables of interest, whereas the aforementioned approaches are not. Our method is also end-to-end differentiable and can be implemented in auto-diff frameworks, welcoming learned components. We are bridging the gap between probabilistic generative models, learning and spatial domain knowledge.

**Depth estimation and differentiable rendering**  Recent promising approaches combine learning and non-parametric categorical distributions for depth estimation (Laidlow et al., 2020; Liu et al., 2019), fusing likelihood terms into a consistent depth estimate. Such depth estimation is compatible with our system and can be used to formulate priors, but for now we rely on a traditional method as a first step (Hirschmuller, 2007). Inferring whole scenes parameterised with neural networks by backpropagation through realistic differentiable rendering has also become a prevalent direction of research (Bi et al., 2020; Mildenhall et al., 2020; Sitzmann et al., 2020). In our method the occupancy and colour map are inferred in a similar way, but the raycasting scheme we follow is simple, meant only to illustrate the framework as a whole. We note that the current inference times of e.g. Bi et al. (2020) amount to days (see appendix of that work), which is hard to scale to online inference. Extending our approach with a more advanced rendering method is the subject of future work.

**Bayesian SLAM inference**  To keep exposition brief, we refer to (Cadena et al., 2016) for an overview of modern SLAM inference and focus only on approaches that have applied fully-Bayesian methods to SLAM. The inference in this work can be categorised as probabilistic SLAM, other prominent examples of which are FastSLAM (Montemerlo et al., 2002) and RBPF SLAM (Grisetti et al., 2005). What distinguishes our method is the application of variational inference with SGVB (Kingma & Welling, 2014). Our model does not restrict the used distributions and allows any differentiable functional form, which enables us to use neural networks. The contribution by Murphy (Murphy, 1999) is one of the first to infer a global map with Bayesian methods. Bayesian Hilbert maps (Senanayake & Ramos, 2017) focus on a fully Bayesian treatment of Hilbert maps for long-term mapping in dynamic environments. Stochastic variational inference is used to infer agent poses from observed 2D image features in (Jiang et al., 2017; Jiang et al., 2019). DVBF-LM (Mirchev et al., 2019) uses Bayes by Backprop (Blundell et al., 2015) for the inference of the global map variable.

## 3  METHOD

**Background**  We adhere to the graphical model of DVBF-LM (Mirchev et al., 2019), but we introduce novel design choices for every model component and implement the overall inference differently, to allow for real-world 3D modelling. In the following, we will first describe the assumed factorisation and then explain the introduced modifications. The assumed joint distribution of all variables is:

$$p(\mathbf{x}_{1:T}, \mathbf{z}_{1:T}, \mathbf{m}_{1:T}, \boldsymbol{\mathcal{M}} \mid \mathbf{u}_{1:T-1})$$
$$= p(\boldsymbol{\mathcal{M}})\rho(\mathbf{z}_1) \prod_{t=1}^{T} p(\mathbf{x}_t \mid \mathbf{m}_t)p(\mathbf{m}_t \mid \mathbf{z}_t, \boldsymbol{\mathcal{M}}) \prod_{t=1}^{T-1} p(\mathbf{z}_{t+1} \mid \mathbf{z}_t, \mathbf{u}_t), \qquad (1)$$

where $\mathbf{x}_{1:T}$ are observations, $\mathbf{z}_{1:T}$ agent states, $\mathbf{m}_{1:T}$ map charts and $\mathbf{u}_{1:T-1}$ conditional inputs (controls). The factorisation defines a traditional state-space model extended with a global map variable $\boldsymbol{\mathcal{M}}$. For a single step $t$, an observation $\mathbf{x}_t$ is generated from a map chart $\mathbf{m}_t$—the relevant extract from the global $\boldsymbol{\mathcal{M}}$ around the current agent pose $\mathbf{z}_t$ (cf. fig. 2a). Chart extraction is given by $p(\mathbf{m}_t \mid \mathbf{z}_t, \boldsymbol{\mathcal{M}})$, which can be seen as an attention mechanism. In this graphical model, SLAM is equivalent to inference of the agent states $\mathbf{z}_{1:T}$ and the map $\boldsymbol{\mathcal{M}}$. For the remainder of this work, we assume all observations $\mathbf{x}_t \in \mathbb{R}^{w \times h \times 4}$ are RGB-D images. Next, we will describe the functional forms of the map $\boldsymbol{\mathcal{M}}$, the attention $p(\mathbf{m}_t \mid \mathbf{z}_t, \boldsymbol{\mathcal{M}})$, the emission $p(\mathbf{x}_t \mid \mathbf{m}_t)$ and the states $\mathbf{z}_{1:T}$.

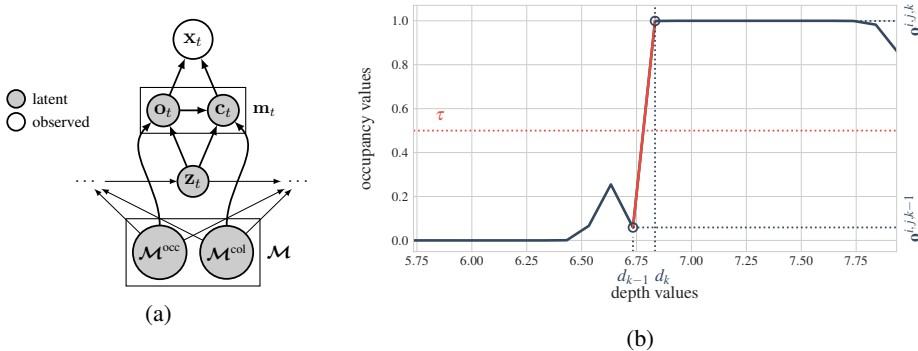

(a)

(b)

Figure 2: (a) One time step of the proposed probabilistic graphical model. (b) Linear interpolation during ray casting for a single ray in the emission model. $d_k$ is the depth corresponding to the first ray value that exceeds $\tau$. The output depth $d$ is formed by linearly interpolating between $d_{k-1}$ and $d_k$ based on the occupancy values $\mathbf{o}^{i,j,k-1}$ and $\mathbf{o}^{i,j,k}$.

**Geometric map**   The map random variable $\boldsymbol{\mathcal{M}} = (\boldsymbol{\mathcal{M}}^{\text{occ}}, \boldsymbol{\mathcal{M}}^{\text{col}})$ consists of two components. $\boldsymbol{\mathcal{M}}^{\text{occ}} \in \mathbb{R}^{l \times m \times n}$ is a spatially arranged 3D grid of scalar values that represent occupancy. $\boldsymbol{\mathcal{M}}^{\text{col}}$ represents the parameters of a feed-forward neural network $f_{\boldsymbol{\mathcal{M}}^{\text{col}}} : \mathbb{R}^3 \rightarrow [0, 255]^3$. The network assigns an RGB colour value to each point in space. In this work, the network weights are deterministic and point-estimated via maximum likelihood, the fully-Bayesian treatment of the colour map is left for future work. The prior and approximate posterior distributions over the occupancy map are:

$$p(\boldsymbol{\mathcal{M}}^{\text{occ}}) = \prod_{i,j,k} \mathcal{N}(\boldsymbol{\mathcal{M}}^{\text{occ}}_{i,j,k} \mid 0, 1), \quad q_\phi(\boldsymbol{\mathcal{M}}^{\text{occ}}) = \prod_{i,j,k} \mathcal{N}(\boldsymbol{\mathcal{M}}^{\text{occ}}_{i,j,k} \mid \mu_{i,j,k}, \sigma^2_{i,j,k}).$$

Here and for the rest of this work $q_\phi$ will denote a variational approximate posterior distribution, with all its optimisable parameters summarised in $\phi$. We assume $p(\boldsymbol{\mathcal{M}}_{\text{occ}})$ and $q_\phi(\boldsymbol{\mathcal{M}}_{\text{occ}})$ factorise over grid cells. The variational parameters $\mu_{i,j,k}, \sigma_{i,j,k}$ are optimised with Bayes by Backprop (Blundell et al., 2015).

**Attention**   In the proposed model, the composition of the attention $p(\mathbf{m}_t \mid \mathbf{z}_t, \boldsymbol{\mathcal{M}})$ and the emission $p(\mathbf{x}_t \mid \mathbf{m}_t)$ implements volumetric raycasting. We engineer them based on our understanding of geometry to ensure generalisation across unseen environments. The attention $p(\mathbf{m}_t \mid \mathbf{z}_t, \boldsymbol{\mathcal{M}})$ forms latent charts $\mathbf{m}_t$, which correspond to extracts from the map $\boldsymbol{\mathcal{M}}$ around $\mathbf{z}_t$. We identify $\mathbf{m}_t$ with the part of the map contained in the frustum of the current camera view. To attend to that region, first the intrinsic camera matrix $\mathbf{K}$ (assumed to be known) and the agent pose $\mathbf{z}_t$ are used to cast a ray for any pixel $[i, j]^T$ in the reconstructed observation. The ray is then discretised equidistantly along the depth dimension into $r$-many points, resulting into a collection of 3D world coordinates $\mathbf{p}_t \in \mathbb{R}^{w \times h \times r \times 3}$. Depth candidate values $d \in \{k\epsilon\}_{1 \leq k \leq r}$ are associated with each point along a ray, where $\epsilon$ is a resolution hyperparameter. The latent chart $\mathbf{m}_t = (\mathbf{o}_t, \mathbf{c}_t)$ factorises into an occupancy chart $\mathbf{o}_t \in \mathbb{R}^{w \times h \times r}$ and a colour chart $\mathbf{c}_t \in \mathbb{R}^{w \times h \times r \times 3}$. Let $p_t^{ijk} \in \mathbb{R}^3$ be a 3D point in the spanned camera frustum. To form the occupancy chart $\mathbf{o}_t$, cells from the map $\boldsymbol{\mathcal{M}}^{occ}$ around $p_t^{ijk}$ are combined with a weighted kernel $o_t^{ijk} = \sum_{l,h,s} \boldsymbol{\mathcal{M}}^{\text{occ}}_{l,h,s} \alpha_{l,h,s}(p^{ijk})$. Note that here $l, h, s$ are indices of the occupancy map voxels. We choose a trilinear interpolation kernel for $\alpha$, merging only eight map cells per point. This makes the attention fast and differentiable w.r.t $\mathbf{z}_t$. The colour chart $\mathbf{c}_t = f_{\boldsymbol{\mathcal{M}}^{\text{col}}}(\mathbf{p}_t)$ is formed by applying $f_{\boldsymbol{\mathcal{M}}^{\text{col}}}$, the colour neural network, *point-wise* to each 3D point. In this work, we keep the chart $\mathbf{m}_t$ deterministic. The full attention procedure can be described as:

$$p(\mathbf{m}_t \mid \mathbf{z}_t, \boldsymbol{\mathcal{M}}) = \prod_{ijk} \delta(\mathbf{m}_t^{ijk} = f_A(\boldsymbol{\mathcal{M}}, p^{ijk})), \quad p^{ijk} = \mathbf{T}(\mathbf{z}_t)\mathbf{K}^{-1}[i, j, 1]^T \underbrace{d}_{:=k\epsilon}.$$

Here $\mathbf{T}(\mathbf{z}_t) \in \mathbb{SE}(3)$ denotes the rigid camera transformation defined by the current agent state $\mathbf{z}_t$ and $i, j, k$ index the points lying inside the attended camera frustum.

**Emission through ray casting**    The emission model factorises over the observed pixels:

$$p(\mathbf{x}_t \mid \mathbf{m}_t) = \prod_{ij} p(\mathbf{x}_t^{ij} \mid \mathbf{m}_t), \quad p(\mathbf{x}_t^{ij} \mid \mathbf{m}_t) = p(d_t^{ij}, \tilde{\mathbf{c}}_t^{ij} \mid \mathbf{o}_t, \mathbf{c}_t).$$

It operates on the extracted chart $\mathbf{m}_t = (\mathbf{o}_t, \mathbf{c}_t)$. Here $\mathbf{x}_t^{ij} \in \mathbb{R}^4$ denotes an RGB-D pixel value, i.e. for each pixel $[i, j]^T$ we reconstruct a depth $d_t^{ij}$ and a colour value $\tilde{\mathbf{c}}_t^{ij}$. The mean of the depth value $d_t^{ij}$ is formed by a function $f_E$:

$$f_E(\mathbf{o}_t)^{ij} = \epsilon \cdot \min_{k \in [r]} k \quad \text{s.t.} \quad \mathbf{o}_t^{ijk} > \tau.$$

$f_E$ traces the ray for pixel $[i, j]^T$, searching for the minimum depth $d = \epsilon k$ for which the occupancy value $\mathbf{o}_t^{ijk}$ exceeds a threshold $\tau$ (a hyperparameter).[1] Since the above $\min$ operation is not differentiable in $\mathbf{o}_t$, we linearly interpolate between the depth value for the first ray hit and its predecessor to form the mean of the emitted depth (cf. fig. 2b):

$$\mu_{d_t}^{ij} = \alpha f_E(\mathbf{o}_t)^{ij} + (1 - \alpha)(f_E(\mathbf{o}_t)^{ij} - \epsilon), \quad \alpha = \frac{\tau - \mathbf{o}_t^{i,j,k-1}}{\mathbf{o}_t^{i,j,k} - \mathbf{o}_t^{i,j,k-1}}.$$

The mean of the emitted colour $\boldsymbol{\mu}_{\tilde{\mathbf{c}}_t}^{ij} = \mathbf{c}_t^{ijk}$ directly corresponds to the $k$-th element of the attended colour values, where $k$ is the index of the first hit from raycasting above. A heteroscedastic Laplace distribution is assumed for both the emitted depth and colour values:

$$p(\mathbf{x}_t^{ij} \mid \mathbf{m}_t) = \text{Laplace}(\mathbf{x}_t^{ij}; (\mu_{d_t}^{ij}, \boldsymbol{\mu}_{\tilde{\mathbf{c}}_t}^{ij}), \text{diag}(\boldsymbol{\sigma}_E^{ij})).$$

**Agent states**    All agent states are represented as vectors $\mathbf{z}_t = (\boldsymbol{\lambda}_t, \boldsymbol{\omega}_t, \mathbf{z}_t^{\text{rest}}) \in \mathbb{R}^{d_{\mathbf{z}}}$. $\boldsymbol{\lambda}_t \in \mathbb{R}^3$ is the agent location in space. $\boldsymbol{\omega}_t \in \mathbb{H}^4$ is the agent orientation, represented as a quaternion. $\mathbf{z}_t^{\text{rest}} \in \mathbb{R}^{d_{\mathbf{z}}-7}$ is a remainder. Depending on the used transition model, $\mathbf{z}_t^{\text{rest}}$ can be $\dot{\boldsymbol{\lambda}}_t$ alone or it can contain an abstract latent portion not explicitly matching physical quantities. The approximate posterior variational family over the agent states factorises over time:

$$q_{\boldsymbol{\phi}}(\mathbf{z}_{1:T}) = \prod_t q_{\boldsymbol{\phi}}(\mathbf{z}_t) = \prod_t \mathcal{N}(\mathbf{z}_t \mid \boldsymbol{\mu}_t^{\mathbf{z}}, \text{diag}(\boldsymbol{\sigma}_t^{\mathbf{z}})^2).$$

Here $\boldsymbol{\mu}_t^{\mathbf{z}} \in \mathbb{R}^{d_{\mathbf{z}}}$ and $\boldsymbol{\sigma}_t^{\mathbf{z}} \in \mathbb{R}^{d_{\mathbf{z}}}$ are free variables for each latent state and are optimised with SGVB (Kingma & Welling, 2014). Notably, the above factorisation over states bears similarity to pose-graph optimisation. One can see the individual terms $q_{\boldsymbol{\phi}}(\mathbf{z}_t)$ as graph nodes, and the loss terms induced by the transition and emission in the objective presented next as the edge constraints.

**Overall objective**    The elements described so far, together with the transition $p(\mathbf{z}_{t+1} \mid \mathbf{z}_t, \mathbf{u}_t)$ discussed in the next section, form the probabilistic graphical model in eq. (1). The assumed variational approximate posterior is

$$q_{\boldsymbol{\phi}}(\mathbf{z}_{1:T}) q_{\boldsymbol{\phi}}(\boldsymbol{\mathcal{M}}) \approx p(\mathbf{z}_{1:T}, \boldsymbol{\mathcal{M}} \mid \mathbf{x}_{1:T}, \mathbf{u}_{1:T-1}).$$

For the optimisation objective we use the negative *evidence lower bound* (ELBO) (Jordan et al., 1999), given as

$$\mathcal{L}_{\text{elbo}} = -\mathbb{E}_q \left[ \sum_{t=1}^{T} \log p(\mathbf{x}_t \mid \mathbf{m}_t) \right]$$
$$+ \text{KL}(q_{\boldsymbol{\phi}}(\boldsymbol{\mathcal{M}}) \,\|\, p(\boldsymbol{\mathcal{M}})) + \mathbb{E}_q \left[ \sum_{t=2}^{T} \text{KL}(q_{\boldsymbol{\phi}}(\mathbf{z}_t) \,\|\, p(\mathbf{z}_t \mid \mathbf{z}_{t-1}, \mathbf{u}_{t-1})) \right]. \quad (2)$$

We employ the approximate particle optimisation scheme from (Mirchev et al., 2019) to deal with long data sequences. The only optimised parameters are $\phi$, containing the parameters of the map and the agent states.

**Making image reconstruction tractable**    Using the full observations during inference is not feasible, as raycasting for all pixels is too computationally demanding. To ensure tractability of the inference method we therefore use reconstruction sampling (Dauphin et al., 2011), emitting a random part of $\mathbf{x}_t$ at a time, by randomly selecting $c$-many pixel coordinates $[i, j]^T$ for every gradient step. Here $c$ is a constant much smaller than the image size $wh$, speeding up gradient updates by a few orders of magnitude. Note that this results in an unbiased, faster and more memory-efficient Monte Carlo approximation of the original objective, avoiding loss of information due to subsampling or sparse feature selection.

---

[1] $\mathbf{o}_t^{ijk}$ is set to 0 for $k \leq 1$ and $f_E(\mathbf{o}_t) = r\epsilon$ if no value exceeds $\tau$ along the ray.

## 4 LEARNING RIGID-BODY DYNAMICS

The introduced model factorisation includes a transition $p(\mathbf{z}_{t+1} \mid \mathbf{z}_t, \mathbf{u}_t)$, which allows the natural inclusion of agent movement priors. This is reflected in the corresponding KL terms in eq. (2). Note that using variational inference lets us integrate any differentiable transition model as-is, without additional linearisation. In the following, we assume the agent has an inertial measurement unit (IMU) providing readings $\ddot{\boldsymbol{\lambda}}_t^{\text{imu}}$ (linear acceleration) and $\dot{\boldsymbol{\omega}}_t^{\text{imu}}$ (angular velocity) over time, which we choose to treat as conditional inputs $\mathbf{u}_t = (\ddot{\boldsymbol{\lambda}}_t^{\text{imu}}, \dot{\boldsymbol{\omega}}_t^{\text{imu}})$.

**Engineering rigid-body dynamics**  In the absence of learning, one can use an engineered transition prior that integrates the IMU sensor readings over time. The latent state $\mathbf{z}_t = (\boldsymbol{\lambda}_t, \boldsymbol{\omega}_t, \dot{\boldsymbol{\lambda}}_t)$ then contains the location, orientation and linear velocity of the agent at every time step. The transition is defined as:

$$p(\mathbf{z}_{t+1} \mid \mathbf{z}_t, \mathbf{u}_t) = \mathcal{N}(\mathbf{z}_{t+1} \mid f_T(\mathbf{z}_t, \mathbf{u}_t), \text{diag}(\boldsymbol{\sigma}_T)^2).$$

The state update $f_T$ implements standard rigid-body dynamics using Euler integration (see appendix D.3). This engineered model will serve as a counterpart for the learned transition model presented next.

**Learning a dynamics model**  Engineered models of the agent movement are often imperfect or not available. We therefore provide a method for learning a fully-probabilistic transition model from streams of prerecorded controls and agent pose observations, which we can then seamlessly include as a prior in the full model. We do not learn the transition with per-step, fully-supervised regression. Instead we formulate a generative sequence model for $T$ time steps. This allows us to separate the aleatoric uncertainty in the observed agent states from the uncertainty in the transition itself. We follow the literature on variational state-space models (Fraccaro et al., 2016; Karl et al., 2017a). We assume we have a sequence of locations $\hat{\boldsymbol{\lambda}}_{1:T}$ and orientations $\hat{\boldsymbol{\omega}}_{1:T}$ as observations, and a sequence of IMU readings, as well as per-rotor revolutions per minute (RPM) and pulse-width modulation (PWM) signals, as conditional inputs $\mathbf{u}_{1:T-1} = (\ddot{\boldsymbol{\lambda}}_{1:T-1}^{\text{imu}}, \dot{\boldsymbol{\omega}}_{1:T-1}^{\text{imu}}, \mathbf{u}_{1:T-1}^{\text{rpm}}, \mathbf{u}_{1:T-1}^{\text{pwm}})$. We define the generative state-space model:

$$p(\hat{\boldsymbol{\lambda}}_{1:T}, \hat{\boldsymbol{\omega}}_{1:T}, \mathbf{z}_{1:T} \mid \mathbf{u}_{1:T-1}) = \delta(\mathbf{z}_1)p(\hat{\boldsymbol{\lambda}}_1, \hat{\boldsymbol{\omega}}_1 \mid \mathbf{z}_1) \prod_{t=1}^{T-1} p_{\boldsymbol{\theta}_T}(\mathbf{z}_{t+1} \mid \mathbf{z}_t, \mathbf{u}_t)p(\hat{\boldsymbol{\lambda}}_{t+1}, \hat{\boldsymbol{\omega}}_{t+1} \mid \mathbf{z}_{t+1}).$$

The objective is to learn generative transition parameters $\boldsymbol{\theta}_T$, such that the marginal likelihood of observed agent poses $p_{\boldsymbol{\theta}_T}(\hat{\boldsymbol{\lambda}}_{1:T}, \hat{\boldsymbol{\omega}}_{1:T} \mid \mathbf{u}_{1:T-1})$ is maximised. The latent state is $\mathbf{z}_t = (\boldsymbol{\lambda}_t, \boldsymbol{\omega}_t, \dot{\boldsymbol{\lambda}}_t, \mathbf{z}_t^{\text{rest}})$, identifying its first three components with location, orientation and linear velocity. The remainder $\mathbf{z}_t^{\text{rest}}$ acts as an abstract state part. Its role is to absorb any quantities that might affect the transition, for example higher moments of the dynamics or sensor biases accumulated over previous time steps. The transition is implemented as a residual neural network on top of Euler integration:

$$p_{\boldsymbol{\theta}_T}(\mathbf{z}_{t+1} \mid \mathbf{z}_t, \mathbf{u}_t) = \mathcal{N}(\mathbf{z}_{t+1} \mid \boldsymbol{\mu}_{t+1}, \text{diag}(\boldsymbol{\sigma}_{t+1})^2)$$

$$\boldsymbol{\mu}_{t+1} = \begin{bmatrix} f_T(\mathbf{z}_t, \mathbf{u}_t) \\ \mathbf{0} \end{bmatrix} + \text{MLP}_{\boldsymbol{\mu}}(\mathbf{z}_t, \mathbf{u}_t), \quad \boldsymbol{\sigma}_{t+1} = \text{MLP}_{\boldsymbol{\sigma}}(\mathbf{z}_t, \mathbf{u}_t),$$

where $f_T$ is the engineered Euler integration from the previous section and the abstract remainder of the latent state is formed entirely by the network (MLP). This strong inductive bias shapes the transition to resemble regular integration in the beginning of training, exploiting engineering knowledge, while still allowing the MLP to eventually take over and correct biases as necessary.

The emission isolates the location and orientation from the latent state as its mean:

$$p(\hat{\boldsymbol{\lambda}}_t, \hat{\boldsymbol{\omega}}_t \mid \mathbf{z}_t) = \mathcal{N}(\hat{\boldsymbol{\lambda}}_t, \hat{\boldsymbol{\omega}}_t \mid (\boldsymbol{\lambda}_t, \boldsymbol{\omega}_t), \text{diag}(\boldsymbol{\sigma})^2).$$

The inference over the latent states uses Gaussian fusion as per (Karl et al., 2017b) and the necessary inverse emission is given by a bidirectional RNN that looks into all observations and conditions:

$$\hat{q}(\mathbf{z}_t \mid \hat{\boldsymbol{\lambda}}_{1:T}, \hat{\boldsymbol{\omega}}_{1:T}, \mathbf{u}_{1:T-1}) = \mathcal{N}(\mathbf{z}_t \mid \text{RNN}(\hat{\boldsymbol{\lambda}}_{1:T}, \hat{\boldsymbol{\omega}}_{1:T}, \mathbf{u}_{1:T-1})).$$

We minimise the negative ELBO w.r.t. $\boldsymbol{\theta}_T$, omitting the conditions in $q$ for brevity:

$$\mathcal{L}(\boldsymbol{\theta}_T) = -\mathbb{E}_q\left[\sum_{t=1}^T \log p(\hat{\boldsymbol{\lambda}}_t, \hat{\boldsymbol{\omega}}_t \mid \mathbf{z}_t)\right] + \mathbb{E}_q\left[\sum_{t=2}^T \mathrm{KL}(q(\mathbf{z}_t) \,\|\, p_{\boldsymbol{\theta}_T}(\mathbf{z}_t \mid \mathbf{z}_{t-1}, \mathbf{u}_{t-1}))\right].$$

## 5 EXPERIMENTS

The experiments are designed to validate three model aspects—the usefulness of the reconstructed 3D world maps, multi-step prediction given future controls and the localisation quality.

For evaluation, we use the Blackbird data set (Antonini et al., 2020). It consists of over ten hours of real quadcopter flight data. The ground truth poses are recorded by a motion capture (MOCAP) system. For each trajectory, Blackbird contains realistic simulated stereo images. We obtain depth from these images using OpenCV's Semi-Global Block Matching (SGBM) (Hirschmuller, 2007) and treat the left RGB camera image and the estimated depth as an observation $\mathbf{x}_t$. We evaluate our method on the test trajectories used in (Nisar et al., 2019). Other trajectories with no overlap are used for training the learned dynamics model and model selection. All model hyperparameters are fixed to the same values for all evaluations. More details can be found in appendices C and D.

### 5.1 DENSE GEOMETRIC MAPPING

A fused dense map, obtained as an approximate variational posterior $q_\phi(\mathcal{M})$, allows us to simulate (emit) the environment from any novel view point. We take the *NYC subway station* Blackbird environment as an example, in which the test set trajectories take place. Figure 3a shows image reconstructions, generated along an example trajectory segment. The model can successfully generate both colour and depth based on $q_\phi(\mathcal{M})$. Note that the true observations are not needed for this, as all of the information is recorded in $\mathcal{M}$ through gradient descent. Even though we use reconstruction sampling during training, on average all image pixels contribute to learning the map. This leads to dense predictions of the agent's sensors. The inferred map correctly filters out wrong observations, as can be seen in the top-down point-cloud comparison in fig. 4d, noting the subway station columns.

### 5.2 USING MAPS FOR DOWNSTREAM TASKS

Besides parameterising predictive emissions, a map can be used to define downstream navigation and exploration tasks. Since the map is modelled as a random variable, the approximate posterior $q_\phi(\mathcal{M})$ also gives us an uncertainty estimate, in this work only applying to $\mathcal{M}^{occ}$. Figure 4a shows a horizontal slice from the occupancy grid, along which the map uncertainty is evaluated. The uncertainty is low inside the subway station, and high on the outside where the agent has not visited. Meaningful map uncertainty is useful for information-theoretic exploration (Mirchev et al., 2019).

The occupancy map can also be used to construct navigation plans, by computing a collision cost-to-go $J(\mathbf{z}_1) = \mathbb{E}_{\mathbf{u}_{1:T}, \mathbf{z}_{1:T} \sim p(\cdot)}[\sum_t c(\mathbf{u}_t, \mathbf{z}_t)]$ (Bertsekas, 2005). For example, the cost $c(\mathbf{u}, \mathbf{z})$ can be the occupancy value in the map at $\mathbf{z}_t$, defining a collision cost. To simulate a control policy, we use an empirical distribution of randomly picked 40-step sequences of IMU readings $\mathbf{u}_{1:T}$ from the test data. Then we generatively sample future states $\mathbf{z}_{1:T}$ for these controls and evaluate the respective costs. Figure 4c shows $J(\mathbf{z})$ evaluated for all states along the considered 2D slice of the map. The cost-to-go is high near the walls and columns, and gradually drops off in the free space.

### 5.3 GENERATING FUTURE PREDICTIONS

The proposed model can predict the agent movement and observations for a sequence of future conditional inputs, i.e. $p(\mathbf{z}_{2:T}, \mathbf{x}_{1:T} \mid \mathbf{u}_{1:T-1}, \mathbf{z}_1)$. Such predictions are crucial for model-based control, and typically not readily available from modern visual SLAM systems. Figure 3b shows predictions of the full spatial model for 200 steps in the future, comparing both the engineered and learned transition priors. Long-term prediction is significantly better with the learned transition, indicating that it corrects biases present in the agent sensors. Conversely, with the engineered transition localisation drift is higher and wrong map regions are queried for generation, translating into wrong depth and colour predictions. We refer to the supplementary video for more examples using the predictive model. Evaluated on 200 test set trajectories with 100 steps, the learned transition

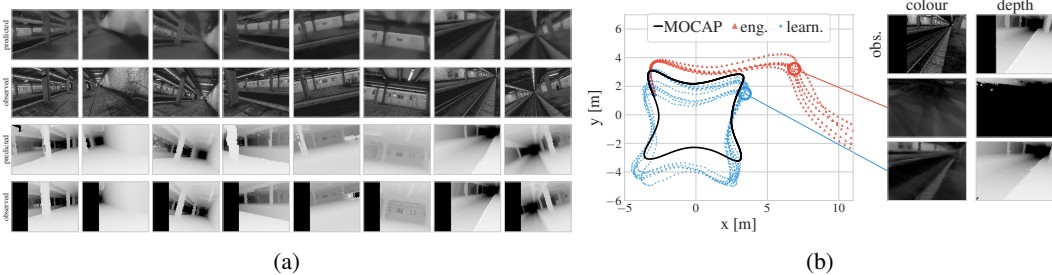

|     |     |
| :-: | :-: |
| (a) | (b) |

Figure 3: (a) Emissions at different trajectory points, sampled at a one second interval. Top to bottom: predicted colour, observed colour, predicted depth, observed depth. (b) Generative predictions using the engineered transition vs. the learned transition in the complete model. Left: top-down view of 200-step location predictions. Right: predicted colour and depth for the same step for both models.

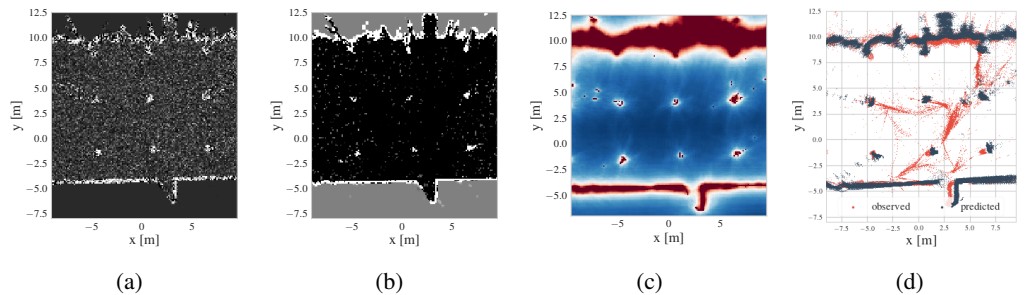

|     |     |     |     |
| :-: | :-: | :-: | :-: |
| (a) | (b) | (c) | (d) |

Figure 4: An illustration of a reconstructed dense map for the *NYC subway station* environment. Note that columns from the subway are captured in the reconstructed map. The figures show a horizontal map slice. (a) Occupancy map uncertainty (white means low uncertainty). (b) Occupancy map mean (white means occupied). (c) Collision cost-to-go (red means high cost). (d) Generated point cloud (black, for inferred agent poses) vs. data point cloud (red, for ground truth MOCAP poses).

leads to better predictions than its engineered counterpart, with an average translational RMSE of $1.156$ and rotational RMSE of $0.096$, compared to $13.768$ and $0.111$ for the engineered model. Similarly, the pixel-wise log-likelihood of the observation predictions is $-1.23$ when the learned model is used, and $-1.85$ for the engineered model, averaged over 1000 images. This evaluation clearly illustrates the positive effect of the learned transition on the model's predictive performance.

## 5.4 AGENT LOCALISATION

Finally, we evaluate localisation performance during SLAM inference. Figure 5 shows estimates for the fastest trajectories in the test set (up to $4m/s$). We refer to appendix F and the supplementary video for more inference examples in different environments. Table 1 summarises the average absolute RMSE for all test trajectories, evaluated following Zhang & Scaramuzza (2018). Here we compare to the results reported by Nisar et al. (2019), including VIMO (Nisar et al., 2019) and VINS-MONO (Qin et al., 2018)—two state-of-the art visual odometry methods. In this case both systems are carefully tuned to the environment. While the RMSE of our method is not as low as that of the baselines, we note that these are absolute values. The online translational error of our method does not exceed $0.4m$, which is practical considering the average $232m$ trajectory length. Our method also succeeds on the fastest *star* test trajectory, for which both baselines have been reported to fail without specific retuning. We note that the two systems run in real-time, whereas currently our method does not—we will tackle real-time inference in our future work (see appendix E).

We also compare to the system benchmarks provided by Antonini et al. (2020), including results for VINS-MONO, VINS-Fusion (Qin et al., 2019) and ORB-SLAM2 (Mur-Artal & Tardós, 2017). We use the same *star* and *picasso* trajectories reported in table 1, and refer to Antonini et al. (2020) for the exact evaluation assumptions. We also explicitly restate a note made by the authors: the benchmarked systems are not carefully tuned to the Blackbird environment and their loop-closure

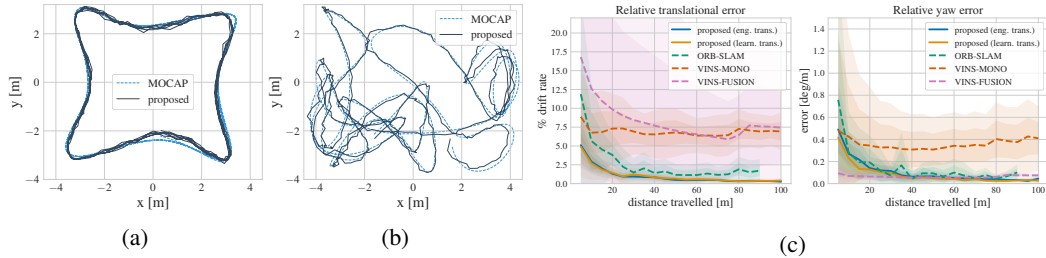

Figure 5: (a,b) Top-down view of localisation for *star, forward yaw, max speed* $4.0m/s$ and *picasso, constant yaw, max speed* $4.0m/s$. (c) Localisation with our method compared to the benchmark results reported by Antonini et al. (2020) for *picasso, constant yaw* and *star, forward yaw*, 1 to $4m/s$.

Table 1: Absolute localisation RMSE for each test trajectory. See appendix B for a discussion of the two noticeable translation RMSE outliers with the learned transition.

|  |  | Translational RMSE [m] | | | | Rotational RMSE [rad] | | | |
|---|---|---|---|---|---|---|---|---|---|
|  |  | proposed (eng. trans.) | proposed (learn. trans) | VIMO | VINS | proposed (eng. trans.) | proposed (learn. trans.) | VIMO | VINS |
| picasso | 1 m/s | 0.139 | 0.143 | 0.055 | 0.097 | 0.053 | 0.052 | 0.013 | 0.011 |
|  | 2 m/s | 0.136 | 0.131 | 0.040 | 0.043 | 0.069 | 0.064 | 0.007 | 0.008 |
|  | 3 m/s | 0.120 | 0.122 | 0.043 | 0.045 | 0.073 | 0.070 | 0.005 | 0.005 |
|  | 4 m/s | 0.174 | 0.368 | 0.049 | 0.056 | 0.124 | 0.149 | 0.009 | 0.011 |
| star | 1 m/s | 0.137 | 0.133 | 0.088 | 0.102 | 0.057 | 0.056 | 0.008 | 0.008 |
|  | 2 m/s | 0.163 | 0.626 | 0.082 | 0.133 | 0.061 | 0.157 | 0.010 | 0.011 |
|  | 3 m/s | 0.281 | 0.187 | 0.183 | 0.235 | 0.080 | 0.059 | 0.015 | 0.016 |
|  | 4 m/s | 0.156 | 0.160 | - | - | 0.065 | 0.059 | - | - |

modules are disabled, which might not be reflective of their best possible performance. Therefore, these baselines represent the performance of an off-the-shelf visual odometry system without changing its hyperparameters. Figure 5c summarises the comparison, reporting error statistics for segments of different length ($x$-axis) divided by the distance travelled. Localisation with our method is robust and drift does not compound, which we attribute to the global map variable $\mathcal{M}$ serving as an anchor.

Overall, localisation is successful for all test trajectories and its accuracy is practical and close to that of state-of-the-art systems, while all merits of deep probabilistic generative modelling are retained.

## 6 CONCLUSION

This work is the first to show that learning a dense 3D map and 6-DoF localisation can be accomplished in a deep generative probabilistic framework using variational inference. The proposed spatial model features an expressive predictive distribution suitable for downstream control tasks, it is fully-differentiable and can be optimised end-to-end with SGD. We further propose a probabilistic method for learning agent dynamics from prerecorded data, which significantly boosts predictive performance when incorporated in the full model. The proposed framework was used to model quadcopter flight data, exhibiting performance close to that of state-of-the-art visual SLAM systems and bearing promise for real-world applications. In the future, we will address the current model's speed limitations and move towards downstream applications based on the learned representation.

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

## A USE OF DOMAIN KNOWLEDGE WHEN DESIGNING LEARNED SPATIAL MODELS

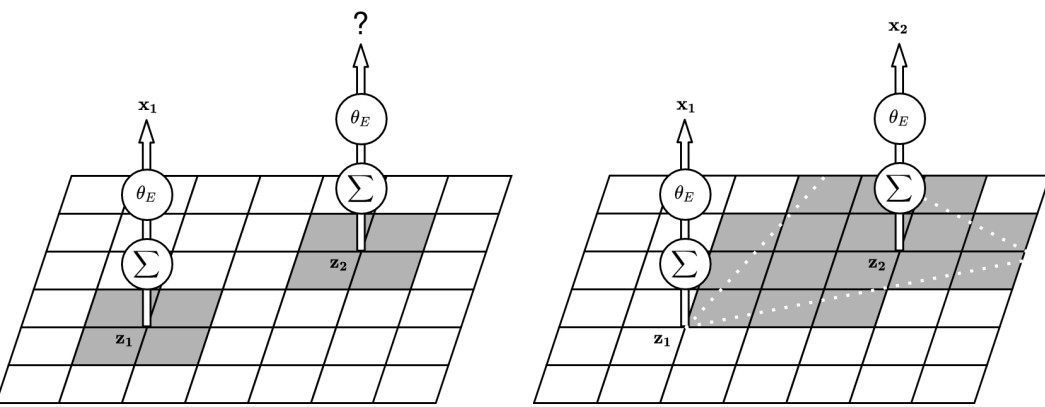

(a) When the attention to the map is too narrow, emitting from observed but yet unvisited places suffers from non-local generalisation issues in an online setting.

(b) A 2D illustration of attending to the whole field of view of the agent. Note that in our method the occupancy map is a 3D volume, and the attended camera frustum resembles a pyramid.

Figure 6: Simplified 2D examples of the conceptual difference the chosen attention can have on generating observations.

What distinguishes our method from DVBF-LM (Mirchev et al., 2019) and other previous 2D and 2.5D deep generative sequence models is that we introduce geometric inductive biases (multiple-view geometry, rigid-body dynamics) in a deep generative sequence model (inferred via the ELBO). We hope the following overview of DVBF-LM's methodology will exemplify why such assumptions could be necessary and motivate our design choices.

DVBF-LM models the world as a 2D chessboard where the content of each grid cell describes a 360° horizontal panoramic view around the z-axis, centered at the agent 2D location. Observation predictions are cropped from that view, relative to the way the agent is facing in 2D (and transformed by an MLP). This already consitutes a set of basic geometric assumptions, but they can be limiting, as explained below. First of all, the agent movement is restricted to a plane, with rotations around one axis (no 6-DoF modelling). The main problem, however, is that the map is updated very locally, only at the 2D location of the agent, because the attention of DVBF-LM only considers four map cells directly underneath. For example, if the agent is standing 5m in front of a wall and facing it, DVBF-LM only stores this information at that location in the map. If the agent moves 2m closer to the wall, it is now accessing different cells, whose content is completely arbitrary (see fig. 6a). These cells will have to learn the presence of the wall again. The map is thus unnecessarily redundant and the agent would have to visit virtually all map cells to infill everything. This means observation predictions from cells which have not been visited before will not be meaningful, even when they were already in the field of view (FoV) of the agent. Localisation also suffers, as the map is more easily allowed to establish multiple modes for the pose posterior because of the redundancy (same content can be stored at different places by mistake), further complicating the perceptual aliasing problem of spatial environments.

By contrast, in our method we use raycasting and attend to the whole camera frustum, accessing all map content in the field of view of the agent, as shown in fig. 6b. Note that the depiction is intentionally simplified to 2D for the sake of clarity, while the actual attention in our approach operates in 3D. In the above example, the wall and the empty space in front of the agent would be captured as soon as the agent sees the wall for the first time, and can be predicted from all regions in the FoV. This constitutes a strong geometric inductive bias that will generalise across different environments.

In general, the lack of geometric assumptions in previous deep spatial generative models (e.g. the ones described in the first paragraph of section 2) is attractive, as it lessens the need for domain knowledge, instead attempting to learn the whole system end-to-end. However, learning everything

becomes problematic (not just in terms of runtime) when the inference needs to happen on-the-fly and data is scarce, as is the case for autonomous agents. When the agent cannot afford to exhaustively observe the whole environment, insufficient data for learning from scratch can lead to problems with consistency in the map, consistency in the long-term predictions and ambiguous agent localisation. One option is to address this through learning certain components in advance (e.g. as we do with the dynamics model, see section 4). Once image data is involved, however, the generalisation of learned components is much harder to ensure because of the curse of dimensionality. Observations can vary greatly from one scene to the next, and free 6-DoF movement of the agent exacerbates the problem further. Instead of collecting data exhaustively, in an attempt to pretrain a network to learn how to render, we rely on the geometric knowledge that is already available to us to design the map, the attention and the emission in section 3. Such assumptions are also at the core of many of the models discussed in the second paragraph in the related work (section 2). In contrast to these models, however, in our work we weave the aforementioned inductive biases into the deep generative framework, which is fully-probabilistic, implies end-to-end differentiability and eases the introduction of new learned components when needed. Therefore, our goal is to maintain an expressive predictive distribution $p(\mathbf{z}_{2:T}, \mathbf{x}_{1:T} \mid \mathbf{u}_{1:T-1}, \mathbf{z}_1)$ that lets us predict the future, precisely aligning with the performed inference based on past data at the same time. We stress that maintaining a full probabilistic predictive model is needed to quantify uncertainty when planning for future actions of the agent.

## B  CASE STUDY: DRONE LANDING

In terms of the integrated transition $p(\mathbf{z}_{t+1} \mid \mathbf{z}_t, \mathbf{u}_t)$, we find that the learned transition performs on par with its engineered counterpart when used for inference in the full model. However, when inspecting the absolute RMSE in table 1, there appear to be two outliers—*picasso, max speed 4.0 m/s* and *star, max speed 2.0 m/s*, when using the learned transition. A closer examination showed that this is entirely due to a large localisation error during landing at the end of the trajectories. In both cases the agent fails to land correctly or falls over, leading to out-of-distribution conditions $\mathbf{u}_t$ from the IMU sensor (cf. fig. 7), for which the learned model does not generalise. Such behaviour is not unexpected when it comes to neural networks. Localisation beforehand is stable along the whole trajectory, the landing tracking failure happens only during the last 2 seconds of movement.

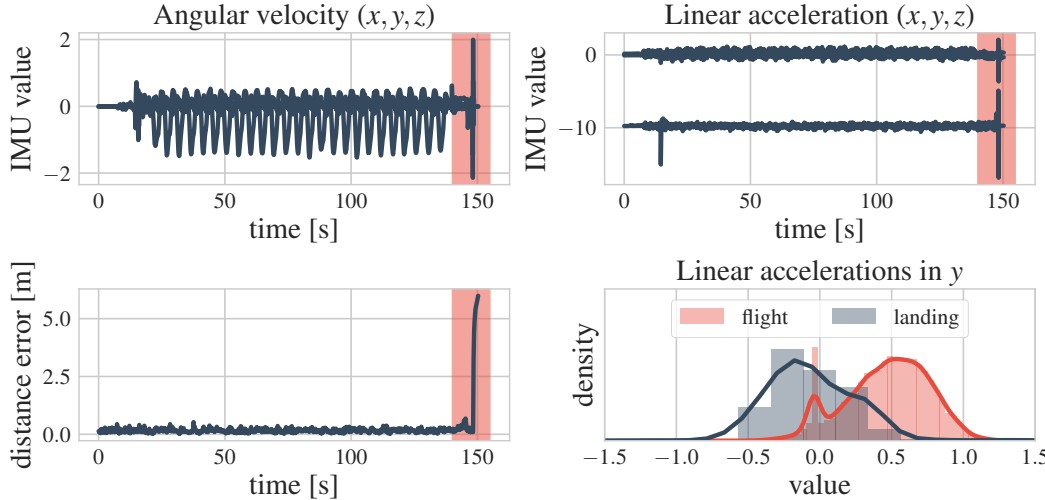

Figure 7: Illustration of the discussed failure landing of the quadcopter at the end of the two test set trajectories, taking *star, max speed 2.0 m/s* as an example. The problematic segment of the trajectory is marked with a red vertical band. Note the outlier controls at the end. Top left: angular velocity IMU readings. Top right: linear acceleration IMU readings. Bottom left: absolute Euclidean distance error w.r.t. the ground truth MOCAP locations. Note how the error is large only at the end of the trajectory, and directly coincides with the outlier controls. Bottom right: comparison of the distribution of linear accelerations in $y$ during flight (red) vs. during landing (gray). The controls during landing are out of distribution for the learned transition model.

# C    DATA DETAILS AND OVERALL SETUP

For all presented experiments we used the Blackbird data set (Antonini et al., 2020), which can be found here: https://github.com/mit-fast/Blackbird-Dataset. In the following we describe the exact pre-processing of the data.

## C.1    DATA PARTITIONING AND USAGE

The data was partitioned into a training, a validation and a test set. The trajectory shapes (e.g. *star*, *picasso*, *patrick*, etc.) in every split were different. This was done to prevent accidental overfitting of the learned transition model to any particular trajectory shape. In particular, the test set contains the trajectories *star, forward yaw* and *picasso, constant yaw*, traversed at different speeds, everything else is used for training and validation. This is the same test setup as the one used by Nisar et al. (2019) (considered as a baseline). Appendix G lists the exact trajectories used for each split, along with the number of steps in each trajectory after pre-processing (which includes subsampling).

The training set was used only to train the learned transition model $p_{\boldsymbol{\theta}_T}(\mathbf{z}_{t+1} \mid \mathbf{z}_t, \mathbf{u}_t)$, following the method described in section 4. The learned transition model is trained separately before it is used as a prior in the full spatial model. Pretraining the model on multiple trajectories beforehand, as opposed to training it from scratch during SLAM inference, ensures that the transition really captures the agent dynamics and does not overfit the currently explored environment. The training trajectories were further randomly rotated around the $z$ axis and linearly translated in space, to promote generalisation.

The validation set was used for checkpointing and selecting the best weights $\boldsymbol{\theta}_T$ of the neural network in the learned transition model, based on the ELBO defined in section 4. The validation set was also used for hyperparameter selection for the engineered transition model, the learned transition model and the full spatial model. The best hyperparameters for all models (cf. appendix D) were selected with random search. You can find details for the search ranges in appendix E.

The test set was used for evaluation only—all results reported in the experimental section of the paper were done on the test data. The full spatial model was tested with the forward yaw *star* trajectories, speeds 1.0 m/s to 4.0 m/s and with the constant yaw *picasso* trajectories, speeds 1.0 m/s to 4.0 m/s, to match the evaluation by Nisar et al. (2019). These trajectories take place in the *NYC subway station* environment.

## C.2    DATA PREPROCESSING

Each trajectory contains IMU, RPM and PWM readings, MOCAP ground truth pose observations, as well as simulated grayscale images from a forward-facing stereo pair and a downward-facing camera.

We pre-processed every trajectory in the following way:

- Ground truth MOCAP state readings were extracted from the provided ROS bags.
- Downward-facing images were ignored.
- The remaining data was subsampled to 10 Hz, using nearest neighbour subsampling based on the provided time stamps.
- Depth was then precomputed from the left and right images using OpenCV's SGBM (Hirschmuller, 2007).
- For every time step, the right colour image was then ignored, while the left image and the depth estimate were together treated as an RGB-D observation $\mathbf{x}_t \in \mathbb{R}^{w \times h \times 4}$.
- The provided IMU readings are measured in the coordinate frame of the IMU sensor. Therefore, they had to be rotated to the body frame of the agent, using the respective quaternion provided with the data set: $\mathbf{q} = [0.707, 0.005, -0.004, 0.706]^T$.

The intrinsic parameters of the camera, specifying the intrinsic camera matrix $\mathbf{K}$, and the stereo baseline were fixed to the values provided with the data set:

- Stereo baseline: 0.1m.
- Image size: $1024 \times 768$.

- Focal length, x: 665.1.
- Focal length, y: 665.1.
- Principal point offset, x: 511.5.
- Principal point offset, y: 383.5.

Depth was computed based on the disparity values produced by SGBM. We used the following parameters, keeping the default values for everything else:

- Block size: 5.
- Min. disparity: 0.
- Max. disparity: 256.

We did not filter the images and the produced depth values in any way and treated them as direct observations, to limit the pre-processing steps as much as possible and rely on the defined spatial model instead. Note that due to the simplicity of the method used for depth estimation, the depth observations contain significant amounts of noise (e.g. fig. 4d). The pixel values for the color images were normalized to be in the range $[0, 1]$.

## D  MODEL DETAILS

This section lists all model and optimisation hyperparameters used for generating the results reported in the paper.

### D.1  FULL SPATIAL MODEL

The full spatial model uses either the learned or the engineered transition, see appendix D.2 and appendix D.3 for their respective hyperparameters.

#### D.1.1  OPTIMISATION

After subsampling, the trajectories in the Blackbird dataset can still contain thousands of time steps. To deal with this, during inference the proposed model follows the approximate particle optimisation scheme introduced by Mirchev et al. (2019), using chunks of 5 time steps for every gradient update, using 50 approximating state particles and refreshing the respective particles after every gradient step.

Adam (Kingma & Ba, 2014) is used to optimise the parameters $\phi$ for the approximate posterior distribution $q_\phi(\mathbf{z}_{1:T}, \mathcal{M})$. Table 2b lists the used optimiser hyperparameters. Note that due to the employed attention $p(\mathbf{m}_t \mid \mathbf{z}_t, \mathcal{M})$, due to the reconstruction sampling in the emission and due to the approximate optimisation scheme mentioned above (which uses only a chunk of the trajectory at a time), only a part of the parameters for the occupancy map and the full trajectory of agent states is used for a gradient step. In other words, gradient updates happen locally in the spatially arranged occupancy map parameters, based on the currently selected local chunk of the agent trajectory. Because of this, momentum is disabled in Adam for both the occupancy map and the agent states, to avoid accidental drift in regions currently not covered by the optimisation.

For every variable in question, the same optimiser is used to optimise the mean and scale of the assumed distribution (where applicable). For any scale distribution parameters (e.g. standard deviations for the assumed Gaussian distributions), the optimisation is performed in log-space to satisfy the positive value constraint.

#### D.1.2  INITIALISATION OF NEW POSES

While performing SLAM, data is added incrementally to the spatial model, to emulate online usage of the method. A new time step $t$ is explored every 500 gradient steps, effectively adding a new observation $\mathbf{x}_t$ and a new conditional input $\mathbf{u}_t$ to the data. Whenever a new time step is added, the parameters of the corresponding state $\mathbf{z}_t$ are unlocked for optimisation and initialized according to table 2a.

Table 2: Model details.

(a) Hyperparameters of the individual spatial model components.

| Component | Parameter | Value |
|---|---|---|
| $\mathbf{z}_t$ | init. value for $\boldsymbol{\mu}_t$ | mean of $p_{\boldsymbol{\theta}_T}(\mathbf{z}_t \mid \boldsymbol{\mu}_{t-1}, \mathbf{u}_t)$ |
| | init. value for $\boldsymbol{\sigma}_t$ | $0.01 \times \mathbf{1}$ |
| $\boldsymbol{\mathcal{M}}^{occ}$ | grid size | $200 \times 200 \times 200$ |
| | init. value for $\mu_{i,j,k}$ | $-0.5$ |
| | init. value for $\sigma_{i,j,k}$ | $0.1$ |
| $\boldsymbol{\mathcal{M}}^{col}$ | # hidden layers | 5 |
| | # hidden units | 256 |
| | activation | softsign |
| | residual connections | true |
| $p(\mathbf{m}_t \mid \mathbf{z}_t, \boldsymbol{\mathcal{M}})$ | $\epsilon$ (ray resolution) | 0.1m |
| | max depth $(k\epsilon)$ | 20m |
| | # reconstr. pixels | 200 |

(b) Optimisation hyperparameters for the full spatial model.

| Variables | Parameter | Value |
|---|---|---|
| $\mathbf{z}_{1:T}$ | Adam, learning rate | 0.001 |
| | Adam, $\beta_1$ | 0.0 |
| | Adam, $\beta_2$ | 0.999 |
| $\boldsymbol{\mathcal{M}}^{occ}$ | Adam, learning rate | 0.05 |
| | Adam, $\beta_1$ | 0.0 |
| | Adam, $\beta_2$ | 0.999 |
| $\boldsymbol{\mathcal{M}}^{col}$ | Adam, learning rate | 0.001 |
| | Adam, $\beta_1$ | 0.9 |
| | Adam, $\beta_2$ | 0.999 |

(c) Hyperparameters of the learned transition model.

| Component | Parameter | Value |
|---|---|---|
| $p_{\boldsymbol{\theta}_T}(\mathbf{z}_{t+1} \mid \mathbf{z}_t, \mathbf{u}_t)$ | # hidden layers | 5 |
| | # hidden units | 64 |
| | activation | relu |
| | residual connections | true |
| | size of $\mathbf{z}^{\text{rest}}$ | 8 |
| $\hat{q}(\mathbf{z}_t \mid \hat{\boldsymbol{\lambda}}_{1:T}, \hat{\boldsymbol{\omega}}_{1:T}, \mathbf{u}_{1:T})$ | RNN type | LSTM |
| | # RNN units | 64 |

### D.1.3 CHOICE OF APPROXIMATE POSTERIOR OVER STATES

In (Mirchev et al., 2019) a bootstrap particle filter is used to implement the variational posterior over agent states. The increased model complexity due to the presented raycasting model makes the application of particle filters with sufficiently many particles costly. In section 3 we therefore introduce an approximate posterior over the agent states $q_\phi(\mathbf{z}_{1:T})$, with free state parameters $\phi$, that factorises over time. The chosen variational states approximation represents a shift in perspective—from filtering towards a method more similar to pose-graph optimisation.

### D.1.4 HANDLING ORIENTATIONS

The portions $\boldsymbol{\omega}_t$ of the sampled latent states $\mathbf{z}_t$ are identified with quaternions and are explicitly normalised to unit quaternions after gradient updates before they are used in the rest of the model. This means that the assumed Gaussian distribution is not directly expressed on the manifold $\mathbb{SO}(3)$, but we found that this parameterisation works well enough.

### D.1.5 COMPONENT PARAMETERS

The rest of the hyperparameters for the full spatial model, including initial values for the optimised variational parameters, are specific to the individual model components and are listed in table 2a.

The scale $\boldsymbol{\sigma}_E$ for the heteroscedastic Laplace emission $p(\mathbf{x}_t \mid \mathbf{m}_t)$ is learned with gradient descent (applied in log-space). The colour part of $\boldsymbol{\sigma}_E$ is forced to be 10 times smaller than that for the depth part, to account for the different scales of the observed values (colour range is $[0, 1]$, depth range is $[0, 20]$).

## D.2 LEARNED TRANSITION

The learned transition is obtained by training the generative parameters $\boldsymbol{\theta}_T$ in the context of the model defined in section 4. When reconstructing orientations in that model, we do not reconstruct quaternions directly because of the ambiguity $\mathbf{q} = -\mathbf{q}$. Instead we reconstruct the rotation matrix corresponding to the observed orientation $\hat{\boldsymbol{\omega}}_t$, i.e. the mean of the emission $p(\hat{\boldsymbol{\omega}}_t \mid \mathbf{z}_t)$ is a rotation matrix constructed from the quaternion $\boldsymbol{\omega}_t$ that's part of the latent state $\mathbf{z}_t$. When used in the full spatial model, the weights of the transition neural network $\boldsymbol{\theta}_T$ are fixed to their pretrained values. This is done to avoid accidental overfitting of the transition parameters to the current trajectory and current environment for which SLAM is performed.

The selected hyperparameters of the learned transition are listed in table 2c.

## D.3 ENGINEERED TRANSITION

The mean of the engineered transition prior $p(\mathbf{z}_{t+1} \mid \mathbf{z}_t, \mathbf{u}_t)$ from section 4 is formed by a function $f_T$, which implements the rigid body dynamics

$$f_T(\mathbf{z}_t, \mathbf{u}_t) = \begin{bmatrix} \boldsymbol{\lambda}_{t+1} \\ \boldsymbol{\omega}_{t+1} \\ \dot{\boldsymbol{\lambda}}_{t+1} \end{bmatrix} = \begin{bmatrix} \boldsymbol{\lambda}_t + \dot{\boldsymbol{\lambda}}_t \Delta t \\ \boldsymbol{\omega}_t \oplus \mathbf{R}(\boldsymbol{\omega}_t) \dot{\boldsymbol{\omega}}_t^{\text{imu}} \Delta t \\ \dot{\boldsymbol{\lambda}}_t + \mathbf{R}(\boldsymbol{\omega}_t) \ddot{\boldsymbol{\lambda}}_t^{\text{imu}} (\Delta t)^2 \end{bmatrix}.$$

Here $\oplus$ denotes standard quaternion integration. The standard deviation $\boldsymbol{\sigma}_T$ of $p(\mathbf{z}_{t+1} \mid \mathbf{z}_t, \mathbf{u}_t)$ (a Gaussian) is a hyperparameter, estimated via search on the validation set. Its diagonal entries are $0.01$ for the location state dimensions, $0.001$ for the orientation state dimensions and $0.001$ for the velocity state dimensions. The gravitational force $\mathbf{g}$ is subtracted from the IMU readings when applying the Euler integration given by $f_T$. The time delta for the integration is set to $\Delta t = 0.1$, to match the 10Hz data subsampling.

# E HYPERPARAMETER SEARCH AND EXECUTION DETAILS

All models are implemented in python using TensorFlow (Abadi et al., 2016), making use of automatic differentiation to optimise model parameters. This allows for the easy integration of neural networks and makes end-to-end optimisation straightforward. In TensorFlow 1.15, one gradient step of the model takes 0.0607s on average, without XLA compilation. In TensorFlow 2.3 one gradient step of the model takes 0.0073s on average, with XLA compilation enabled. The inference experiments in the paper currently assume 500 iterations per added data point (once every 0.1s for a 10Hz stream), which amounts to 36.5s of inference runtime per 1s of real-time movement in the newer TensorFlow version. As already mentioned in the main text, we have not yet optimised the model for real-time inference, and we will address this in our future work. The large speed-up achieved by simply enabling XLA and switching to a higher version of TensorFlow indicates that a lot of the computations in the model can be improved further (e.g. by moving to a C++ runtime or writing dedicated CUDA kernels). We also anticipate that better initialisation and careful tuning of the model hyperparameters will let the system reach interactive operation rates.

All probabilistic modelling aspects were implemented using Edward (Tran et al., 2018). Hyperparameter search (HPS) experiments were executed in a cluster with 8 Tesla V100 GPUs and 40 Intel Xeon E5-2698 CPU cores. Because of the large amount of Blackbird data (4.7 TB) and the current model's speed limitations, the search was performed for a subset of all possible hyperparameters. The performed model selection is therefore not exhaustive, possibly leaving potential for improvement. In the following we list the considered hyperparameter search ranges and the number of trials for each model.

## E.1 HPS: FULL SPATIAL MODEL

The hyperparameters for the full spatial model were selected based on localisation RMSE for 500-step segments of trajectories in the validation set. A total of 200 experiments were conducted, randomly picking a trajectory segment on which SLAM inference is performed. The considered parameter ranges are listed in table 3a.

Table 3: HPS details.

(a) HPS ranges for the full spatial model.

| Parameter | Range |
|---|---|
| reconstr. pixel count | $[100, 200, 500]$ |
| $q(\boldsymbol{\mathcal{M}}^{occ})$ init. value stddev | $[0.01, 0.1, 1.0]$ |
| $p(\boldsymbol{\mathcal{M}}^{occ})$ stddev | $[0.1, 1.0, 10.0]$ |
| $\boldsymbol{\mathcal{M}}^{occ}$ grid side | $[50, 100, 200]$ |
| $\boldsymbol{\mathcal{M}}^{occ}$ learning rate | $[0.01, 0.05, 0.001]$ |
| $\boldsymbol{\mathcal{M}}^{col}$ learning rate | $[0.01, 0.05, 0.001]$ |
| $\mathbf{z}_{1:T}$ learning rate | $[0.01, 0.05, 0.001]$ |

(b) HPS ranges for the learned transition model.

| Parameter | Range |
|---|---|
| learning rate | $[0.0001, 0.0005, 0.001, 0.005]$ |
| $\mathbf{z}^{\text{rest}}$ | $[8, 16]$ |
| # hidden units | $[32, 64, 128, 256]$ |
| # layers | $[2, 3, 4, 5]$ |
| activation | $[\text{softsign}, \text{relu}]$ |
| # RNN units | $[32, 64, 128, 256]$ |

(c) HPS ranges for the engineered transition model.

| Parameter | Range |
|---|---|
| $\boldsymbol{\lambda}$ stddev | $[0.0001, 0.001, 0.01]$ |
| $\boldsymbol{\omega}$ stddev | $[0.0001, 0.001, 0.01]$ |
| $\dot{\boldsymbol{\lambda}}$ stddev | $[0.0001, 0.001, 0.01]$ |

### E.2 HPS: LEARNED TRANSITION MODEL

The learned transition hyperparameters were selected based on the validation set ELBO value from the model discussed in section 4. A total of 400 experiments were conducted, using all of the blackbird training and validation data. The considered parameter ranges are listed in table 3b.

### E.3 HPS: ENGINEERED TRANSITION MODEL

The engineered transition hyperparameters were selected in the same HPS for the full model discussed above, based on localisation RMSE for 500-step segments of trajectories in the validation set. The considered parameter ranges are listed in table 3c.

# F FURTHER INFERENCE EXAMPLES

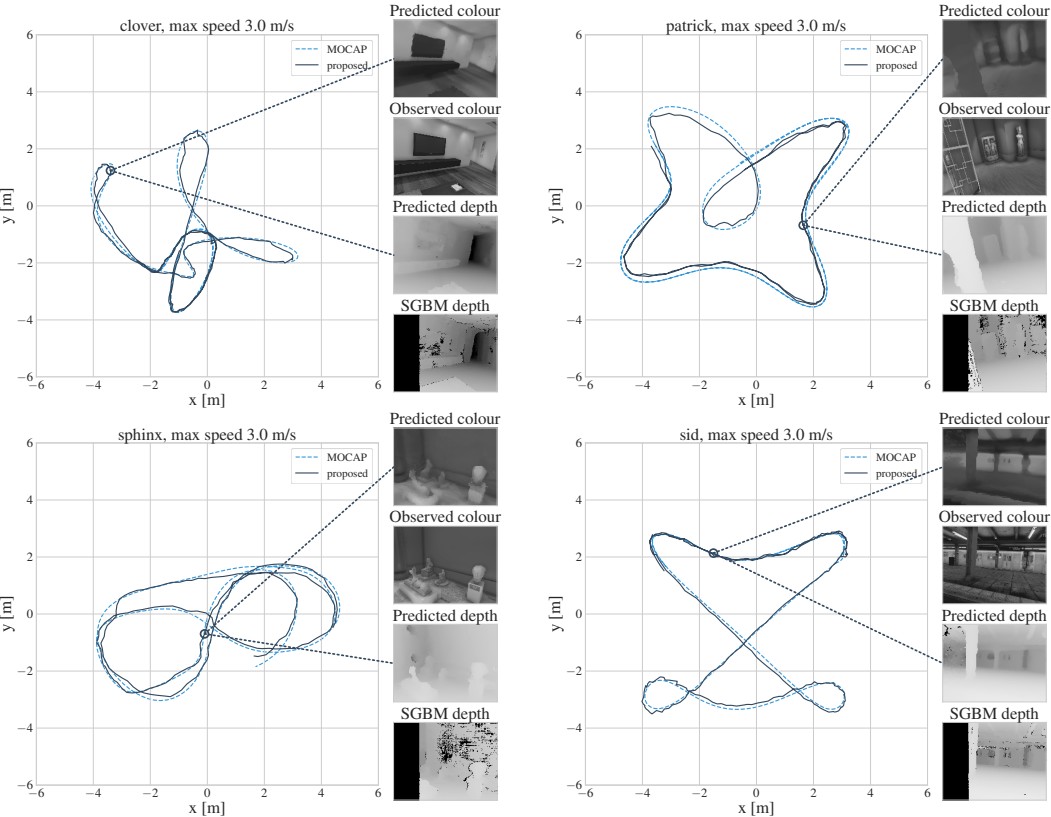

Figure 8: Examples of inference in the full spatial model, expressed in localisation and mapping for segments of the following trajectories: *clover, forward yaw, max speed 3.0 m/s*, *patrick, forward yaw, max speed 3.0 m/s*, *sphinx, forward yaw, max speed 3.0 m/s* and *sid, forward yaw, max speed 3.0 m/s*. Left on every subfigure: top-down view of the localisaton estimate. Right on every subfigure: depth and colour reconstructions from the generative model for one time step.

# G  DATA SET SPLITS

| Training set | | |
|---|---|---|
| trajectory | # steps | duration [s] |
| 3dFigure8, constant yaw, 1.0 m/s | 2087 | 208.7 |
| 3dFigure8, constant yaw, 2.0 m/s | 2130 | 213.0 |
| 3dFigure8, constant yaw, 3.0 m/s | 2091 | 209.1 |
| 3dFigure8, constant yaw, 4.0 m/s | 2136 | 213.6 |
| 3dFigure8, constant yaw, 5.0 m/s | 2240 | 224.0 |
| ampersand, constant yaw, 1.0 m/s | 2086 | 208.6 |
| ampersand, constant yaw, 2.0 m/s | 2061 | 206.1 |
| ampersand, constant yaw, 3.0 m/s | 833 | 83.3 |
| clover, constant yaw, 1.0 m/s | 2575 | 257.5 |
| clover, constant yaw, 2.0 m/s | 2605 | 260.5 |
| clover, constant yaw, 3.0 m/s | 833 | 83.3 |
| clover, constant yaw, 4.0 m/s | 2631 | 263.1 |
| clover, constant yaw, 5.0 m/s | 2607 | 260.7 |
| dice, constant yaw, 2.0 m/s | 2605 | 260.5 |
| dice, constant yaw, 3.0 m/s | 833 | 83.3 |
| dice, constant yaw, 4.0 m/s | 2614 | 261.4 |
| figure8, constant yaw, 1.0 m/s | 1513 | 151.3 |
| figure8, constant yaw, 2.0 m/s | 2052 | 205.2 |
| figure8, constant yaw, 5.0 m/s | 1999 | 199.9 |
| mouse, constant yaw, 1.0 m/s | 2641 | 264.1 |
| mouse, constant yaw, 2.0 m/s | 2674 | 267.4 |
| mouse, constant yaw, 3.0 m/s | 833 | 83.3 |
| mouse, constant yaw, 4.0 m/s | 2620 | 262.0 |
| mouse, constant yaw, 5.0 m/s | 2655 | 265.5 |
| oval, constant yaw, 2.0 m/s | 2033 | 203.3 |
| oval, constant yaw, 3.0 m/s | 833 | 83.3 |
| oval, constant yaw, 4.0 m/s | 2035 | 203.5 |
| thrice, constant yaw, 1.0 m/s | 2726 | 272.6 |
| thrice, constant yaw, 2.0 m/s | 2658 | 265.8 |
| thrice, constant yaw, 3.0 m/s | 2656 | 265.6 |
| thrice, constant yaw, 4.0 m/s | 2656 | 265.6 |
| thrice, constant yaw, 5.0 m/s | 2621 | 262.1 |
| tiltedThrice, constant yaw, 1.0 m/s | 2624 | 262.4 |
| tiltedThrice, constant yaw, 2.0 m/s | 2624 | 262.4 |
| tiltedThrice, constant yaw, 3.0 m/s | 833 | 83.3 |
| tiltedThrice, constant yaw, 4.0 m/s | 2602 | 260.2 |
| tiltedThrice, constant yaw, 5.0 m/s | 2574 | 257.4 |
| winter, constant yaw, 1.0 m/s | 2625 | 262.5 |
| winter, constant yaw, 2.0 m/s | 2570 | 257.0 |
| winter, constant yaw, 3.0 m/s | 833 | 83.3 |
| winter, constant yaw, 4.0 m/s | 2569 | 256.9 |
| winter, constant yaw, 5.0 m/s | 2620 | 262.0 |
| ampersand, forward yaw, 1.0 m/s | 1100 | 110.0 |
| ampersand, forward yaw, 2.0 m/s | 893 | 89.3 |
| clover, forward yaw, 1.0 m/s | 1088 | 108.8 |
| clover, forward yaw, 2.0 m/s | 1088 | 108.8 |
| clover, forward yaw, 3.0 m/s | 833 | 83.3 |
| clover, forward yaw, 4.0 m/s | 1101 | 110.1 |
| clover, forward yaw, 5.0 m/s | 1091 | 109.1 |
| dice, forward yaw, 1.0 m/s | 1096 | 109.6 |
| dice, forward yaw, 2.0 m/s | 1099 | 109.9 |
| dice, forward yaw, 3.0 m/s | 833 | 83.3 |
| mouse, forward yaw, 1.0 m/s | 1110 | 111.0 |
| mouse, forward yaw, 2.0 m/s | 1107 | 110.7 |
| mouse, forward yaw, 3.0 m/s | 833 | 83.3 |
| mouse, forward yaw, 4.0 m/s | 1108 | 110.8 |
| mouse, forward yaw, 5.0 m/s | 1107 | 110.7 |
| oval, forward yaw, 1.0 m/s | 1086 | 108.6 |
| oval, forward yaw, 2.0 m/s | 892 | 89.2 |
| oval, forward yaw, 3.0 m/s | 833 | 83.3 |
| oval, forward yaw, 4.0 m/s | 1086 | 108.6 |
| thrice, forward yaw, 1.0 m/s | 1088 | 108.8 |
| thrice, forward yaw, 2.0 m/s | 1088 | 108.8 |
| thrice, forward yaw, 3.0 m/s | 833 | 83.3 |
| thrice, forward yaw, 4.0 m/s | 1088 | 108.8 |
| thrice, forward yaw, 5.0 m/s | 1088 | 108.8 |
| tiltedThrice, forward yaw, 1.0 m/s | 898 | 89.8 |
| tiltedThrice, forward yaw, 2.0 m/s | 1088 | 108.8 |
| tiltedThrice, forward yaw, 3.0 m/s | 833 | 83.3 |
| tiltedThrice, forward yaw, 4.0 m/s | 1088 | 108.8 |
| tiltedThrice, forward yaw, 5.0 m/s | 1087 | 108.7 |
| winter, forward yaw, 2.0 m/s | 1089 | 108.9 |
| winter, forward yaw, 3.0 m/s | 833 | 83.3 |
| winter, forward yaw, 4.0 m/s | 1091 | 109.1 |

| Test set | | |
|---|---|---|
| trajectory | # steps | duration [s] |
| star, constant yaw, 1.0 m/s | 2680 | 268.0 |
| star, constant yaw, 2.0 m/s | 2635 | 263.5 |
| star, constant yaw, 3.0 m/s | 2640 | 264.0 |
| star, constant yaw, 4.0 m/s | 2667 | 266.7 |
| star, constant yaw, 5.0 m/s | 2611 | 261.1 |
| star, forward yaw, 1.0 m/s | 1491 | 149.1 |
| star, forward yaw, 2.0 m/s | 1503 | 150.3 |
| star, forward yaw, 3.0 m/s | 1523 | 152.3 |
| star, forward yaw, 4.0 m/s | 1627 | 162.7 |
| star, forward yaw, 5.0 m/s | 1108 | 110.8 |
| picasso, constant yaw, 1.0 m/s | 2040 | 204.0 |
| picasso, constant yaw, 2.0 m/s | 2071 | 207.1 |
| picasso, constant yaw, 3.0 m/s | 2072 | 207.2 |
| picasso, constant yaw, 4.0 m/s | 2109 | 210.9 |
| picasso, constant yaw, 5.0 m/s | 2626 | 262.6 |
| picasso, forward yaw, 1.0 m/s | 833 | 83.3 |
| picasso, forward yaw, 3.0 m/s | 2083 | 208.3 |
| picasso, forward yaw, 4.0 m/s | 2057 | 205.7 |
| picasso, forward yaw, 5.0 m/s | 891 | 89.1 |

| Validation set | | |
|---|---|---|
| trajectory | # steps | duration [s] |
| sid, constant yaw, 1.0 m/s | 2688 | 268.8 |
| sid, constant yaw, 2.0 m/s | 2677 | 267.7 |
| sid, constant yaw, 3.0 m/s | 833 | 83.3 |
| sid, constant yaw, 4.0 m/s | 2668 | 266.8 |
| sid, constant yaw, 5.0 m/s | 2661 | 266.1 |
| sid, forward yaw, 1.0 m/s | 897 | 89.7 |
| sid, forward yaw, 2.0 m/s | 1109 | 110.9 |
| sid, forward yaw, 3.0 m/s | 833 | 83.3 |
| sid, forward yaw, 4.0 m/s | 1109 | 110.9 |
| sid, forward yaw, 5.0 m/s | 1110 | 111.0 |
| sphinx, constant yaw, 1.0 m/s | 2612 | 261.2 |
| sphinx, constant yaw, 2.0 m/s | 2601 | 260.1 |
| sphinx, constant yaw, 3.0 m/s | 833 | 83.3 |
| sphinx, constant yaw, 4.0 m/s | 2560 | 256.0 |
| sphinx, forward yaw, 1.0 m/s | 1088 | 108.8 |
| sphinx, forward yaw, 2.0 m/s | 1087 | 108.7 |
| sphinx, forward yaw, 3.0 m/s | 833 | 83.3 |
| sphinx, forward yaw, 4.0 m/s | 1086 | 108.6 |
| bentDice, constant yaw, 1.0 m/s | 2624 | 262.4 |
| bentDice, constant yaw, 2.0 m/s | 2698 | 269.8 |
| bentDice, constant yaw, 3.0 m/s | 417 | 41.7 |
| bentDice, constant yaw, 4.0 m/s | 2632 | 263.2 |
| bentDice, forward yaw, 1.0 m/s | 1088 | 108.8 |
| bentDice, forward yaw, 2.0 m/s | 1088 | 108.8 |
| bentDice, forward yaw, 3.0 m/s | 833 | 83.3 |
| patrick, constant yaw, 1.0 m/s | 2598 | 259.8 |
| patrick, constant yaw, 2.0 m/s | 2584 | 258.4 |
| patrick, constant yaw, 3.0 m/s | 417 | 41.7 |
| patrick, constant yaw, 4.0 m/s | 2611 | 261.1 |
| patrick, constant yaw, 5.0 m/s | 2580 | 258.0 |
| patrick, forward yaw, 1.0 m/s | 1089 | 108.9 |
| patrick, forward yaw, 2.0 m/s | 1089 | 108.9 |
| patrick, forward yaw, 3.0 m/s | 833 | 83.3 |
| patrick, forward yaw, 4.0 m/s | 1091 | 109.1 |

