# OpenReview forum: "Variational State-Space Models for Localisation and Dense 3D Mapping in 6 DoF"
_ICLR.cc/2021/Conference — ICLR 2021 Poster_

### Official Review · AnonReviewer2 · 2020-10-28
**Good formulation of dense RGB-D SLAM as a deep state-space model, but what are the advantages?**

**Rating:** 6
**Confidence:** 4

**Review:**

Summary

This paper addresses the problem of dense RGB-D SLAM. The key idea is to formulate the problem as a deep state-space model and infer the state of the latent variables (i.e. pose and geometry) using variational inference. The experiments demonstrate that the method performs well in a challenging quadcopter dataset. However, the advantages of the approach are not demonstrated.

Strengths
- The proposed approach is a very elegant and principled formulation of the dense SLAM problem. The probabilistic graphical model considers the observations, dynamics and latent states of the agent (i.e. the pose and dense map). As far as I am aware, not many deep slam approaches consider all these factors in the estimation process. The proposed model is therefore a nice unification of neural networks, dynamics and multi-view geometry.

- The method has a number of interesting capabilities such as the ability to predict the future trajectory of the camera. The demonstration of the usefulness of the inferred map in downstream tasks such as navigation is also very interesting.

Weaknesses
- The related work is severely lacking. Specifically, the paper seems to marginalise most of the recent methods that try to combine deep learning with dense SLAM. The authors summarize the whole collection of such approaches with a single sentence: “depth or semantic representations with existing SLAM methods has also become a prevalent direction of research”. This isn’t a fair representation of existing works and doesn’t really do much to position the authors’ novel contribution with respect to the existing literature. The authors need to discuss and contrast their approach to works such as BA-Net, DeepFactors , Neural RGB->D, DeepSFM etc. Currently, the related work is too focussed on traditional SLAM approaches (like VinsMono) and 2D/2.5D methods like VAST/DVBF-LM.

- The proposed method requires RGB-D as input which is obtained from SGM in the paper. How sensitive is the approach to noise in the depth input, and can the approach work with just RGB input? The reason I consider this a weakness is that many existing RGB-D SLAM approaches (even traditional ones like ORB-SLAM2) have very high accuracy and in some sense can be considered a “solved problem”. This is not really the case for dense monocular SLAM which is still a very challenging problem.

- Furthermore, the advantages of the proposed approach are not described in the paper. In the evaluation the improvement over ORB-SLAM2 in terms of accuracy doesn’t appear to be very significant. Is it simply an improvement in accuracy or are there other advantages as well?

- The experiments and comparisons are also rather limited. The authors have only evaluated on a single quadcopter dataset which doesn’t really show the generalizability of the trained model. ORB-SLAM2 and VINS-MONO don’t have any trained components and are therefore completely general (apart from the need for tuning some settings). How does the proposed approach perform when trained on Blackbird and evaluated on EuRoC, for example?

- The runtime of SLAM approaches is quite important, especially for application on platforms such as quadcopters. It would be good if the authors could include some indication of the runtime and whether the inference can be done in realtime.

Minor points

It would be good to bold the best performing methods in the table as this makes it easier to see the relative performance of the methods.

---

> ### Author Response · Authors · 2020-11-17
> **Reply**
>
> Thank you for your time and feedback. In the following we address your comments one by one.
>
> > Furthermore, the advantages of the proposed approach are not described in the paper. In the evaluation the improvement over ORB-SLAM2 in terms of accuracy doesn’t appear to be very significant. Is it simply an improvement in accuracy or are there other advantages as well?
>
> Thank you for acknowledging that future predictions are interesting and our inferred maps appear useful. We consider this to be a major advantage of our system compared to visual SLAM SOTA methods like ORB-SLAM2. We discussed this in more detail in our overall response to all reviewers. In short, it wasn’t our sole goal to outperform these systems in localization accuracy, but rather to establish a consistent framework with a generative model useful for control tasks that aligns with the inference.
>
> > The related work is severely lacking.
>
> Thank you for the provided references, we have included all of them in our related work (primarily in subsections two and three). We hope you will find it appropriate.
>
> Here we briefly note that approaches like DeepFactors, BA-Net and DeepSFM share a lot of the motivation behind our method, as they combine domain knowledge from traditional SLAM with learning, for the sake of generalization and real-world modelling. However, similar to traditional SLAM methods, these methods also primarily focus on inference and do not explicitly maintain the predictive distribution needed for control. This is the main distinction compared to our method.
>
> > The proposed method requires RGB-D as input which is obtained from SGM in the paper. How sensitive is the approach to noise in the depth input, and can the approach work with just RGB input?
>
> In our supplementary video, there are quite a few examples of noisy depth observations (e.g. from 0:47 to 1:33). You can observe this in the panel that depicts the observed depth, as well as in the top-down view of the data point cloud (orange) on the right. Apart from that, figure 4d shows the same plot for one of the test trajectories (data point cloud is in red). In general, the SGBM depth observations were far from ideal. Despite that, localization on the test trajectories in the paper and on the trajectories in the supplementary video was stable.
>
> Working with monocular RGB data would technically be possible in our framework, but would require careful initialization of the occupancy and treating the generated depth after the emission model as a latent variable.
>
> > The experiments and comparisons are also rather limited. The authors have only evaluated on a single quadcopter dataset which doesn’t really show the generalizability of the trained model.
>
> The pretrained drone dynamics model is the only model component that would need to generalize across environments. Since the learned dynamics model is not conditioned on images, its generalization only depends on the physics of the agent itself.
>
> We do not expect the learned transition to generalize to different drones and vehicles, as the movement physics of another agent will inevitably differ. In terms of our evaluation, we explicitly made sure that the dynamics model is learned from trajectory data which is very different from the trajectories we used for testing (e.g. see Appendix G). We note that the Blackbird data set offers a lot of variability in terms of the trajectory shapes and the flight speeds of the drone.
>
> We will address different data sets with different agents as soon as we have improved the inference runtime of our model.
>
> > The runtime of SLAM approaches is quite important, especially for application on platforms such as quadcopters.
>
> We added a more detailed discussion of the current inference runtime in the beginning of Appendix E and we comment on this in our overall response. We note that we have not specifically tuned our model for real-time use, and we will address this in our future work.

---

### Official Review · AnonReviewer1 · 2020-10-29
**A novel learning based SLAM algorithm, but slower and less precise than SOTA methods**

**Rating:** 6
**Confidence:** 5

**Review:**

This paper presents a novel learning-based visual-inertial odometry algorithm. The algorithm simultaneously reconstructs the world map as well as the states of the agent from the stereo RGBD sensors.  The world is modeled as an occupancy grid with color. A graphical model with attention mechanism and ray casting is used to model how the world and the agent state renders the RGBD sensor data. ELBO is used to optimize the model. The technical details look sound to me.

The learning part of the paper is to use neural networks to model the dynamics: the distribution of the next agent state given the current state and the IMU is modeled as a normal distribution whose mean and standard deviation are determined via a neural network that takes the current state and the IMU as the input. The neural network is trained on pre-recorded agent states and IMU data captured in the MOCAP. Note that the neural network builds upon the Euler integration based IMU model; it models the residual between the true agent next state and the prediction using the Euler-integration based IMU model.

The paper performs extensive experiments on a dataset that includes agents of different velocities and different types of trajectories. Qualitatively the trajectories generated by the proposed method look great in the figures. Quantitatively, the average localization error is around 0.2% of the trajectory length, demonstrating the practicality of the method.

However, compared with the SOTA methods, the proposed method is slower and at the same time has higher localization errors. In addition, the learning-based dynamics model does not outperform the Euler integration based IMU model, despite the former builds upon the latter. The paper also reports two outlier data points in the experiments where the learning-based dynamics model leads to significantly higher localization errors. In the appendix, the authors analyze these two data points and identify the cause: the landing of the agents in these two trials fail and lead the model to learn ungeneralizable dynamics.

The presentation of the paper is great. The technical parts of the paper are written concisely and precisely with clear motivation and intuition. I find the paper easy to understand.

To summarize:

Pros:
1. Great presentation
2. Novel formulation and learning-based dynamics model
3. Extensive experiments

Cons:
1. The proposed method is behind the state-of-the-art in both speed and localization accuracy
2. The learning-based dynamics model proposed by the model does not outperform the closed-form Euler integration based one

---

> ### Author Response · Authors · 2020-11-17
> **Reply**
>
> Thank you for your time and feedback. In the following we will address your comments one by one.
>
> > This paper presents a novel learning-based visual-inertial odometry algorithm.
>
> We wanted to reiterate that in addition to inference (visual-inertial odometry, also inferring a fused map), our approach also offers a predictive distribution $p(z_{2:T}, x_{1:T} \mid z_1, u_{1:T-1})$, which can act as a simulator of the environment. Please consider the common response to all reviewers for more details.
>
> > However, compared with the SOTA methods, the proposed method is slower and at the same time has higher localization errors.
>
> Thank you for acknowledging that a relative localization error of around 0.2% of the trajectory length is still practical. We agree, the accuracy of SOTA visual SLAM systems is still higher than that of our approach, and such methods are generally real-time capable. We have addressed this in more detail in our overall response.
>
> Given that our formulation has not been explored previously, we hope that eventually our method will reach the same level of maturity. We also note figure 5c, which shows that even established SLAM systems are often prone to failure if not carefully tuned, and that the trade-off between robustness and accuracy is still an open question.
>
> > In addition, the learning-based dynamics model does not outperform the Euler integration based IMU model, despite the former builds upon the latter.
>
> We are afraid there might be a misunderstanding. The learned dynamics model outperforms the engineered model in terms of predictive performance as can be seen in the last 40 seconds of the provided supplementary video and in section 5.3 in the paper. It is true that during inference the learned transition does not appear more accurate, which is because the inverse measurement / emission part can correct both the engineered and learned dynamics, as expected in a SLAM context
>
> > The paper also reports two outlier data points in the experiments where the learning-based dynamics model leads to significantly higher localization errors. (...) the landing of the agents in these two trials fail and lead the model to learn ungeneralizable dynamics.
>
> The failure cases of the learned transition during drone landing occur because it has never been trained on such data, only at the very end of the trajectories. I.e. it does not generalize to that data only. This would be a problem of most learning approaches with flexible models such as MLPs or GPs, which are not expected to fare well when their inputs at test time are out-of-distribution w.r.t. the training data. Except for these two landing examples, the learned dynamics work well on the test set trajectories.
>
> > The presentation of the paper is great. The technical parts of the paper are written concisely and precisely with clear motivation and intuition. I find the paper easy to understand.
>
> We are very happy. If there is anything that you believe can be improved, in terms of making our motivation clearer, please let us know.

---

### Official Review · AnonReviewer3 · 2020-10-29
**ICLR 2021**

**Rating:** 6
**Confidence:** 3

**Review:**


## Summary

The paper proposes a framework built on DVBF-LM, extended to dense 3D mapping. Overall I find the work somewhat incremental over DVBF-LM and that the methodology lacks clarity.

## Strengths

 - The work builds on a fundamentally new and interesting line of generative variational approaches to SLAM
 - It also builds on recent work in differentiable SLAM systems
 - It shows results on datasets taken from an actual quadrotor.
 - Results seem promising (but comparisons to some other related approaches might be missing)


## Weaknesses

 - I found the methodology extremely difficult to follow. There are many variables used, and some choices are simply not explained or not clear. This becomes very laborious. One such example is that the variable $o$ is used to represent occupancy as well as the agent location in space. In all these variables, I missed a definition for $q_\phi$. Part of this is also that non-standard terms and notation are used. As far as I know, "map charts" is not a common term. Neither is emission model (should be measurement model). Although I do see that these notations are borrowed from DVBF-LM.

 - I feel there are some related works that are missing from the discussion. The approach in [1] formulates the fully differentiable dense SLAM problem (including ray casting) on real world datasets, [2] is a recent novel 3D rendering approach and also does differentiable ray casting,  [3] demonstrates an end to end approach to learning measurement likelihood models with an RL active localization framework, and [4] which is also uses a clever combination of deep learning and multi-view geometry to produce dense 3D maps. There are others. In general, differentiable ray casting is not new, and the authors should make a more concerted effort to cite this work and place their approach within this context.

[1] ∇ SLAM: Dense SLAM meets Automatic Differentiation
KM Jatavallabhula, G Iyer, L Paull
2020 IEEE International Conference on Robotics and Automation (ICRA), 2130-2137

[2] Neural Reflectance Fields for Appearance Acquisition
S Bi, Z Xu, P Srinivasan, B Mildenhall, K Sunkavalli, M Hašan, ...
arXiv preprint arXiv:2008.03824

[3] Active Neural Localization
DS Chaplot, E Parisotto, R Salakhutdinov
6th International Conference on Learning Representations (ICLR-18)

[4] Towards the Probabilistic Fusion of Learned Priors into Standard Pipelines for 3D Reconstruction
T Laidlow, J Czarnowski, A Nicastro, R Clark, S Leutenegger
2020 IEEE International Conference on Robotics and Automation (ICRA), 7373-7379

[5] Neural scene representation and rendering
SMA Eslami, DJ Rezende, F Besse, F Viola, AS Morcos, M Garnelo, ...
Science 360 (6394), 1204-1210

 - It would nice to know in the related work section what exactly it is about previous methods that makes them not applicable to "real-world 3D modelling". In particular, the very related DVBF-LM which method builds upon.

 - The parts of the "Method" section which are taken from DVBF-LM should be moved to a section called "Background." That would enable the reader to more clearly differentiate what is new here.

## Other Comments/Questions

 - You state that " $M^col$ represents the parameters of a feed-forward neural network $f_{M^col}: R3 \mapsto [0, 255]^3$. The network assigns an RGB colour value to each point in space." What colours are assigned to free space? What about viewpoint dependent illumination?

---

> ### Author Response · Authors · 2020-11-17
> **Reply**
>
> Thank you for your review and feedback. In the following we will address your comments one by one.
>
> > The parts of the "Method" section which are taken from DVBF-LM should be moved to a section called "Background."
>
> Most of the “Method” section consists of advances over DVBF-LM. Only the first paragraph lists the common factorization of the graphical model, which we share with DVBF-LM.
>
> The definition of the occupancy and color map, the attention, the rendering emission, the 6-DoF states, the state inference, the reconstruction sampling and the learned dynamics model are all novel and necessary improvements over DVBF-LM. These are required to move from 2.5DoF to 6DoF, as listed in the “Introduction”. Please let us know if anything concerning the distinction remains unclear.
>
> > What exactly is it about previous methods that makes them not applicable to "real-world 3D modelling". In particular, the very related DVBF-LM.
>
> Previous literature specifically on deep sequential generative models has been mostly applied to planar scenarios with 2D agent movement. This manifests itself e.g. in the choice of a grid for a map representation instead of a full 3D volume, or in the state of the agent being limited to 2D movement and discrete rotations (vs. 6DoF).  In the case of DVBF-LM, this is largely because the attention model does not consider the field of view of the agent and the observation model is an abstract MLP. We counteract this by explicitly introducing geometric inductive biases (e.g. differentiable raycasting). The first subsection of the revised related work and specifically the new Appendix A discuss this in more detail.
>
> We very much value the predictive distribution that comes with such generative models, as it is a prerequisite for optimal control of the agent. We largely attribute scalability issues in previous generative spatial systems to the lack of stronger geometric priors. In the case of spatial observations (images, lidars) it is hard to generalise such methods across different environments. In particular, online inference where data is scarce can be challenging.
>
> We also do not find that all existing works that deal with spatiality and learning are limited to 2D simulated environments. E.g. almost all of the methods with geometric assumptions we refer to in the second and third subsections of our related work already scale to 3D scenes.
>
> > I feel there are some related works that are missing from the discussion.
>
> Thank you for the provided references. We agree that they help to better position our method in the field. We used them to extend our related work section. Hopefully you will find it appropriate.
>
> In summary, our system is not intended as a replacement for existing realistic differentiable rendering like in [2] or depth estimation methods like in [4]. In fact, they synergize with our framework and could replace the simpler counterparts we currently use. Yet, the inference times in [2] amount to days. This makes the integration of such renderers in systems like ours that are meant to eventually run in real-time challenging.
>
> The main point of our model is that it establishes a probabilistic generative predictive distribution for $p(z_{2:T}, x_{1:T} \mid z_1, u_{1:T-1})$ while also scaling it up to 3D. Methods like the above do not address this. While [3,5] also offer a predictive distribution (although [5] does not model the agent dynamics), these have only been applied to 2D / 2.5D environments. In [1] differentiability in SLAM is approached in a different way than we do, e.g. the gradients through the rendering procedure flow from the map to the observations. This is useful for determining the contribution of observations during map fusion, but for learning a map via gradient descent the opposite gradients are needed.
>
> > I found the methodology extremely difficult to follow. There are many variables used, and some choices are simply not explained or not clear.
>
> We are sorry if the notation in the paper led to confusion. We have corrected both points in the uploaded manuscript revision. As you already noted, “map chart” was already established in DVBF-LM and we did not want to deviate for consistency. We believe “emission” is an established term for the model between the latent states and the observations in Hidden Markov Models (HMMs).
>
> In terms of explaining our design choices better, we completely revised our related work section and we added a more detailed explanation of our motivation in Appendix A.
>
> > What colours are assigned to free space? What about viewpoint dependent illumination?
>
> Free space is governed by the occupancy network in our model, the color network only needs to predict at the points in space which are occupied. The network outputs in the free space are not controlled for, as they do not have any meaning in the graphical model. Considering viewpoint-dependent illumination is left for future work (also consider the comment on inference times in [2] above).

---

### Official Review · AnonReviewer4 · 2020-11-02
**A good paper - accept.**

**Rating:** 7
**Confidence:** 4

**Review:**

This paper describes a Deep Variational Bayes Filter (DBVF) for Deep-Learning based SLAM in 3D environments. It builds upon similar work for 2D environments in [Mirchev et. al. 19], and learns a full 3D RGBD occupancy map and a sequence of 6 DoF poses (localization) using raw stereoscopic camera data. Differentiable ray-casting and an attention model is described to access the learnt global map to give a local map and an expected observation - using an emission model from the current pose and local map. A transition model describing the evolution of the dynamics of the agent is also learnt. A variational approximation of the actual posterior (of the sequence of poses and the map, given the sequence of observations) is learnt by optimizing the standard ELBO equation from Variational Bayes. Such deep generative models, once learnt (in an unsupervised way) for an environment, allows one to hallucinate a sequence of poses and observations, given the learnt map and control inputs. This allows downstream robotic control tasks like environment exploration and path planning to be integrated into the model. Experiments on a simulated dataset with a flying drone in a subway and living room environments demonstrate good SLAM performance (that approach traditional methods): bird's eye view projections of the 6 DoF poses and the emitted maps closely match the ground truth poses and the occupancy grid. This is a well-written paper that represents a good step up from [Mirchev et. al. 19] to formulate a DVBF with realistic RGBD data streams. The authors mention that the computational times for this method is still far from conventional SLAM techniques - an actual quantification of the time taken during inference would be useful.

---

> ### Author Response · Authors · 2020-11-17
> **Reply**
>
> Thank you for your positive review and the provided feedback!
>
> > The authors mention that the computational times for this method is still far from conventional SLAM techniques - an actual quantification of the time taken during inference would be useful.
>
> Appendix E now has a more detailed discussion of the current inference runtime. As stated in the paper conclusion, it is a main priority of ours to get the model inference to run in real-time, as that will enable optimal control and planning on real hardware.

---

### Author Response · Authors · 2020-11-17
**Overall response**

We thank all reviewers for the taken time and the constructive feedback, it is much appreciated.

We would like to address common points that were raised and better clarify our motivation. The responses to each of our reviewers contain more thorough discussions of the individual questions.

**What are the advantages relative to the visual SLAM SOTA?**

Visual SLAM SOTA systems do not feature predicting the future. For model-based optimal control, both a state estimator and a predictive distribution are needed, but current systems deliver only the former. Let us go into more detail w.r.t. these two aspects.

*Prediction*

We want to emphasize that our model is generative, with parametric models for the agent dynamics, for a fused dense map and for rendering complete observations. This lets us define $p(z_{2:T}, x_{1:T} \mid z_1, u_{1:T-1})$. This approach has recently been referred to as a world model, and is a fundamental block needed for model-based optimal control [1]. it enables the agent to predict the consequences of its actions. We illustrate this in sections 5.2, 5.3 and in the last 40 seconds of our supplementary video.

We have made an effort to ensure that:
- The predictive distribution is aligned with the inference (ELBO objective).
- The predictive distribution is expressive (e.g. predicting whole future observations).
- The predictive distribution quantifies uncertainty (important for control).

This improves over current visual SLAM systems.

*Inference (i.e. SLAM)*

We acknowledge that the localization inference accuracy of our model comes close to, but does not outmatch the best visual SLAM models. However, this was not the sole focus of our work, as already explained. Traditional SLAM is a field with decades of historical development and tuning, we will be working on improving the inference accuracy of our method in the future.

Note that bundle adjustment [2], the de facto visual SLAM inference method, implies the reprojection of one image frame into the next (also known as warping). This is auto-regressive in nature, and often done for a sparse set of image points. In contrast, the assumptions of our model explicitly target the rendering of individual observations (without dependence on previous frames, due to the fused map) and the inference conforms to that. This enables the predictive model we just described.

**What are the advantages to previous spatial learning models?**

There are two main streams of work that we believe should be considered. We have done our best to address this better in the related work in our new revision.

Our method attempts to scale previous deep generative state-space models, previously explored in simulated 2D and 2.5D environments, to realistic 3D modelling and free agent movement. We do this by providing sufficient domain knowledge to the system (in the form of differentiable raycasting, occupancy mapping, etc.). In doing so, we also preserve the favorable properties of deep generative modelling (generation from a latent space, fully-probabilistic, end-to-end differentiable).

On the other hand, a number of recent works combine learning with spatial domain knowledge, targeting real-world scene representation, depth estimation and SLAM. To the best of our knowledge fully-probabilistic generative models have not been considered in that context yet. This is the gap we are trying to bridge---combining generative modelling, learning and domain knowledge from multiple-view geometry and robotics.

**Inference runtime**

We’ve added a discussion in Appendix E. The current execution times are far from real-time, ca. 35 times too slow. We believe this is mostly due to inefficiencies of the python runtime, the automatic-differentiation framework (TensorFlow), insufficient tuning of hyperparameters, no careful initialization of the map and new states added to the system. It is our main priority to address this in our future work, as it will enable experiments with the system on real hardware. But it is clearly out of the scope of the current work–we estimate it to fill 1-2 follow-up papers.

**Change log**

- Significantly expanded the related work. We have included discussions and comparisons to all previous works provided by our reviewers.
- Added a new appendix section, Appendix A, discussing the motivation behind our model assumptions in more detail.
- Added a discussion of the inference runtime in Appendix E.
- Improved the clarity of the notation.

#### References
[1] Dimitri P. Bertsekas. Dynamic Programming and Optimal Control, volume I. Athena Scientific, Belmont, MA, USA, 3rd edition, 2005.

[2] Lucas, B.D. and Kanade, T., 1981. An iterative image registration technique with an application to stereo vision.

---

### Decision · Program_Chairs · 2021-01-07
**Final Decision**

**Decision:**

Accept (Poster)

**Comment:**

The paper proposes a method for SLAM like dense 3D mapping (colored occupancy grid) based on differentiable rendering with a possibility to provide a probabilistic generative predictive distribution, evaluated on UAVs.

Initially this paper has a wide spread of reviews, with ratings between 4 and 9. Reviewers appreciated the elegant and principled formulation and the interest of the predictive distribution. On the downside, several issues were raised on the incremental nature wrt to DVBF-LM; presentation and writing being very dense and difficult to follow; positioning wrt to prior art; performance with respect to known visual SLAM SOTA baselines; limited evaluations.

The authors provided responses to many of this questions and also updates to the paper, which convinced several reviewers, who unanimously recommended acceptance after discussion.

The AC concurs.